# CAR/CXCR5-T cell immunotherapy is safe and potentially efficacious in promoting sustained remission of SIV infection

Mary S. Pampusch[1][◎], Hadia M. Abdelaal[1][◎], Emily K. Cartwright[1], Jhomary S. Molden[1], Brianna C. Davey[1], Jordan D. Sauve[1], Emily N. Sevcik[1], Aaron K. Rendahl[1], Eva G. Rakasz[2], Elizabeth Connick[3], Edward A. Berger[4], Pamela J. Skinner[1]*

1 Department of Veterinary and Biomedical Sciences, University of Minnesota, St. Paul, Minnesota, United States of America, 2 Wisconsin National Primate Research Center, University of Wisconsin, Madison, Wisconsin, United States of America, 3 Division of Infectious Diseases, University of Arizona, Tucson, Arizona, United States of America, 4 Laboratory of Viral Diseases, NIAID, NIH, Bethesda, Maryland, United States of America

◎ These authors contributed equally to this work.

* skinn002@umn.edu

**Data Availability Statement:** All relevant data are within the manuscript and its supporting information files.

## Abstract

During chronic human immunodeficiency virus (HIV) or simian immunodeficiency virus (SIV) infection prior to AIDS progression, the vast majority of viral replication is concentrated within B cell follicles of secondary lymphoid tissues. We investigated whether infusion of T cells expressing an SIV-specific chimeric antigen receptor (CAR) and the follicular homing receptor, CXCR5, could successfully kill viral-RNA⁺ cells in targeted lymphoid follicles in SIV-infected rhesus macaques. In this study, CD4 and CD8 T cells from rhesus macaques were genetically modified to express antiviral CAR and CXCR5 moieties (generating CAR/CXCR5-T cells) and autologously infused into a chronically infected animal. At 2 days post-treatment, the CAR/CXCR5-T cells were located primarily in spleen and lymph nodes both inside and outside of lymphoid follicles. Few CAR/CXCR5-T cells were detected in the ileum, rectum, and lung, and no cells were detected in the bone marrow, liver, or brain. Within follicles, CAR/CXCR5-T cells were found in direct contact with SIV-viral RNA⁺ cells. We next infused CAR/CXCR5-T cells into ART-suppressed SIV-infected rhesus macaques, in which the animals were released from ART at the time of infusion. These CAR/CXCR5-T cells replicated in vivo within both the extrafollicular and follicular regions of lymph nodes and accumulated within lymphoid follicles. CAR/CXR5-T cell concentrations in follicles peaked during the first week post-infusion but declined to undetectable levels after 2 to 4 weeks. Overall, CAR/CXCR5-T cell-treated animals maintained lower viral loads and follicular viral RNA levels than untreated control animals, and no outstanding adverse reactions were noted. These findings indicate that CAR/CXCR5-T cell treatment is safe and holds promise as a future treatment for the durable remission of HIV.

**Funding:** This study was supported by NIH grants (https://www.nih.gov/grants-funding) 5R01AI096966-06S1 (PS, EC, and EB), 1UM1AI26617 (PS, EC and EB), P51OD011106/ P51RR000167 (ER), MN REACH grant 5U01HL127479-03 (PS), R01AI143380 (PS and EB), 1UM14126617 (PS and EC), training grant 5-T32 AI 55433 (EKC) as well as funds provided by the NIAID Division of Intramural Research (https://www.niaid.nih.gov) and the NIH Intramural AIDS Targeted Antiviral Program (https://irp.nih.gov) (EB). The funders had no role in study design, data collection and analysis, decision to publish and preparation of the manuscript.

**Competing interests:** We have read the journal's policy and the authors of this manuscript have the following competing interests: Pamela Skinner is the co-founder and CSO of MarPam Pharma and has a patent pending US20180371057A1. Mary Pampusch was a former employee of MarPam Pharma. Other authors declare that no competing interests exist.

## Author summary

A person infected with human immunodeficiency virus (HIV) has replicating virus concentrated within the follicles of lymphoid tissues. The cells needed to clear the infection, cytotoxic T lymphocytes, have limited access to follicles and, thus, the cytotoxic T lymphocytes are never completely able to clear all of the HIV from the body. In this study, we have produced immunotherapeutic T cells that home to follicles and clear infected cells. These T cells express a viral targeting chimeric antigen receptor (CAR) and a molecule called CXCR5, which leads to homing of the cells to follicles. Upon administration of these CAR T-cells to virus-infected primates, we found that the cells localized to the follicle, replicated, and directly interacted with infected cells. While the cells were not maintained in the animals for more than 4 weeks, most of the treated animals maintained lower levels of virus in the blood and follicles than untreated control animals. This study shows that this immunotherapy has potential as a treatment leading to long-term remission of HIV without the need for antiretroviral drugs.

## Introduction

Over 37 million people worldwide live with human immunodeficiency virus (HIV) [1]. Antiretroviral therapy (ART) effectively reduces levels of viral replication in patients; however, ART is not a cure, as it fails to fully eliminate the cellular reservoir of the virus [2–4]. Successful control of HIV-1 requires life-long adherence to ART, which can be challenging for patients with limited or intermittent access to healthcare. Treatment fatigue, side effects [5], and inconsistent access to medications have led to unsatisfactory levels of treatment adherence that range from 27–80% across various populations, in a disease requiring 95% adherence to be effective [6]. Alarmingly, only 57% of people living with HIV (PLWH) in the US are virally suppressed with ART [7], contributing to the public health threat, as untreated PLWH can transmit virus to others. Increasingly, such transmission can lead to development of drug-resistant strains of HIV [8–10]. To improve the health of individuals with HIV and to reduce community transmission, there is intense global interest in fully eradicating HIV or in developing a strategy for durable remission in the absence of ART [11–17].

Chimeric antigen receptor (CAR)-T cells have shown great promise in the treatment of certain cancers, including B cell leukemia and B cell lymphomas [18–22]. CAR-T cells are of particular interest in the treatment of HIV, as T cells can be engineered to specifically target the envelope glycoprotein expressed on the surface of HIV-infected cells. In this study, we employed a CAR that targets two independent regions of the viral glycoprotein gp120 [12,23]; the CAR contains CD4 (domains 1 and 2) and the carbohydrate recognition domain of mannose-binding lectin (MBL). The MBL prevents CD4 from acting as a viral entry receptor, the bispecific antigen recognition increases the anti-viral potency of the CAR, and the use of self domains minimizes the possibility of immunogenicity [23].

During chronic HIV and simian immunodeficiency virus (SIV) infections prior to AIDS progression, the vast majority of viral replication concentrates within B cell follicles [24–30], primarily within T follicular helper cells (Tfh) [24–26,31,32]. Free virions in immune complexes are also localized in follicles through binding to follicular dendritic cells (FDC) in germinal centers (GC) [33–41]. By contrast, levels of virus-specific $CD8^+$ T cells, which are critical in controlling HIV and SIV infections, are found at relatively low levels within B cell follicles [26,27,42,43]. In fact, we previously reported the average ratio of in vivo effector (virus-specific $CD8^+$ T cells) to target (virus-infected cells) cells is over 40-fold lower in

follicular (F) compartments compared to extrafollicular (EF) compartments of secondary lymphoid tissues [27]. Further, we reported that levels of SIV in F areas during early infection [44] and levels of SIV in both F and EF areas during chronic infection are inversely correlated with levels of virus-specific CD8[+] T cells in these compartments [27]. In addition, we found that levels of virus-specific CTL in lymphoid tissues correlate with reductions in viral loads [45,46]. Migration of lymphocytes into B cell follicles is directed by the binding of the chemokine receptor, CXCR5 [47–49] to the chemokine ligand, CXCL13 [50–52], which is produced by follicular stromal cells, such as marginal reticular cells and FDC, and by GC Tfh cells [53–59]. Thus, expression of CXCR5 on the surface of a CD8+ T cell mediates migration into B cell follicles [60,61].

We hypothesize that increasing the numbers of HIV-specific CD8[+] T cells in B cell follicles will decrease viral replication within the follicles and lead to durable control of HIV [62,63]. In this study, we tested this hypothesis in an SIV-infected rhesus macaque model of HIV infection. Infusion of SIV-infected rhesus macaques (both ART-naïve and ART-suppressed) with rhesus-specific (CD4-MBL) CAR/CXCR5-T cells showed preliminary evidence that the treatment was safe and potentially efficacious in promoting sustained remission of SIV infection. After infusion, CAR/CXCR5-T cells proliferated, accumulated in B cell follicles, interacted with viral (v) RNA[+] cells, and showed an association with reduced levels of follicular vRNA[+] cells and overall decrease in plasma viral load. These studies indicate that CAR/CXCR5-T cell immunotherapy shows promise as a tool in the development of durable remission of HIV infection without the need for life-long ART.

## Results

### CAR/CXCR5-T cells home to lymphoid follicles and contact SIV-infected cells in vivo

To evaluate the localization and relative abundance of CAR/CXCR5-T cells in lymphoid and non-lymphoid tissues, and the relative localization within lymphoid tissues of CAR/CXCR5-T cells and SIV vRNA[+] cells, we autologously infused CAR/CXCR5-transduced T cells into an SIVmac239-chronically infected rhesus macaque and sacrificed the animal 2 days post-treatment (DPT). The transduced T cell product was labeled with the fluorescent dye Cell Trace Violet (CTV), and infused into the animal at a dose of $0.35 \times 10^8$ cells/kg. Ten minutes post-infusion, the infused CTV positive cells made up 0.17% of the lymphocyte population and 42% of that population expressed MBL and CXCR5. At necropsy, the CTV positive cells made up 0.055% of the PBMC population. Spleen, lymph node (LN), rectum, ileum, bone marrow, lung, liver, and brain were collected at 2 DPT, and examined for localization of the CTV-labeled cells by confocal microscopy (Fig 1). The infused CTV-labeled T cells accumulated in both F and EF areas of the spleen and LN, with a few cells detected in the rectum, and lung. No infused CTV-labeled T cells were detected in ileum, bone marrow, liver, or brain tissue. Whilst we did not detect CTV labeled cells in the fresh ileum tissue sections evaluated, these sections did not contain lymphoid aggregates. We did a follow-up assay with RNAscope to determine whether CAR/CXCR5-T cells were present in Peyer's patches and other lymphoid aggregates within the ileum and found a few CAR/CXCR5-T cells located within lymphoid aggregates (S1 Fig).

To evaluate the localization of CAR/CXCR5-T cells relative to SIV vRNA[+] cells, we used duplex RNAScope in situ hybridization (ISH) [64–66], which allows simultaneous detection of both gammaretroviral vector-transduced CAR/CXCR5-T cells and SIV-infected cells. These assays detect RNA, and we presume most vRNA[+] and CAR/CXCR5[+] cells detected with RNAscope are expressing viral proteins and CAR and CXCR5 proteins, respectively.

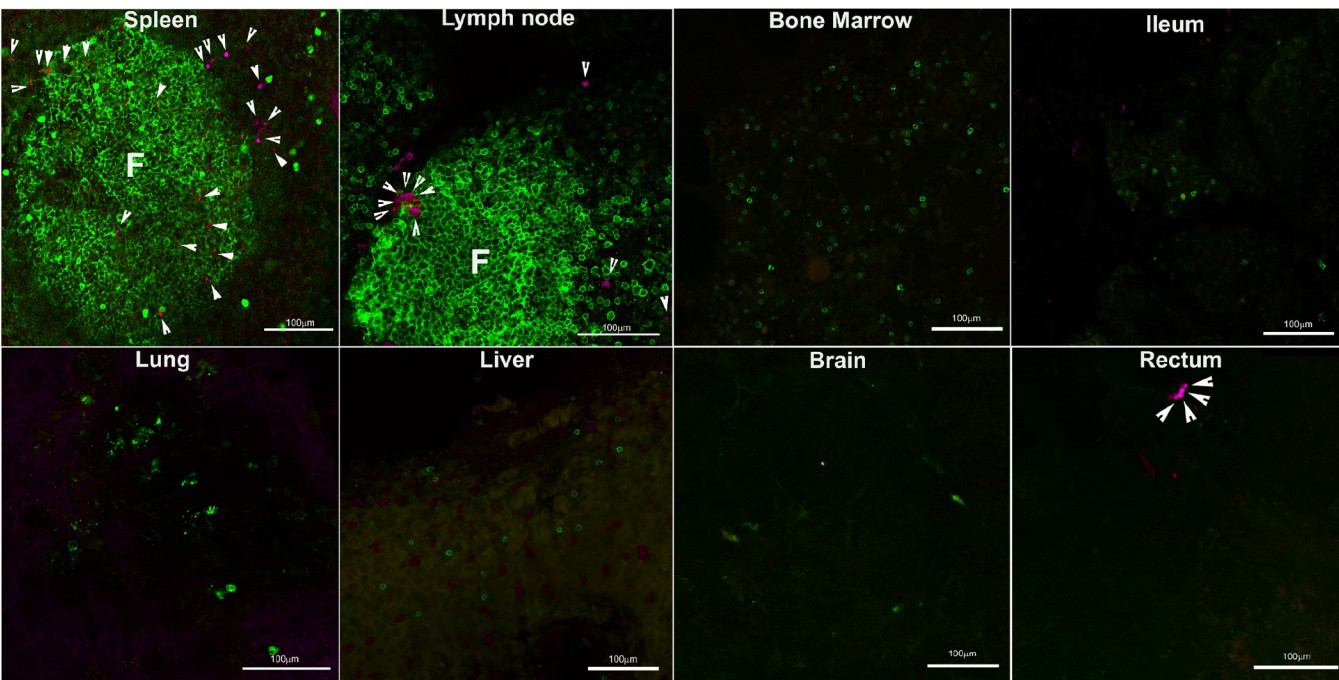

**Fig 1. Distribution of CTV-labeled infused T cell product in different tissues at 2 days post-treatment.** Location of cell trace violet (CTV)-labeled infused T cells (pseudo-colored magenta) was determined in a chronically SIV-infected ART-naïve rhesus macaque, animal R14025 at two days post-treatment (DPT). CTV-labeled infused T cell product was detected in both the follicular and extrafollicular areas of lymphoid tissues. Representative images from spleen, lymph node (LN), bone marrow, ileum, lung, liver, brain, and rectum. Tissues were stained with anti-IgM or anti-CD20 (green) to label B cells and delineate B cell follicles (green). Arrowheads point to CTV-labeled cells. Scale bars = 100 μm.

Spleen tissue sections were hybridized to two sets of probes, one that specifically binds the gammaretroviral CAR/CXCR5 construct and another that specifically binds SIV vRNA (Fig 2). In addition to detecting SIV vRNA$^+$ cells, SIV virions trapped by the follicular dendritic cell (FDC) network were detected as a white haze within the B cell follicle as described previously [60,65]. The duplex RNAScope ISH was combined with immunofluorescence staining to allow the delineation of F and EF areas in lymphoid tissues. CAR/CXCR5-T cells were detected primarily in F areas of the spleen (50 cells/mm$^2$ in F compared to 1.7 in EF) leading to an E:T of CAR/CXCR5-T cells: vRNA$^+$ cells of 3 and 1.4 in F and EF areas respectively. In some instances, CAR/CXCR5-T cells were detected in direct contact with SIV vRNA$^+$ cells (Fig 2). CAR/CXCR5-T cells need to migrate to and directly contact SIV vRNA$^+$ target cells to provide death-inducing signals via a cytotoxic immunological synapse. We found 3.3% (23/689) of CAR/CXCR5-T cells were in direct contact with SIV vRNA$^+$ cells, and 3.7% (23/621) of SIV vRNA$^+$ cells were in direct contact with CAR/CXCR5-T cells. We further examined SIV vRNA$^+$ cells to determine the distance between the virally infected cells and the nearest CAR/CXCR5-T cell. The median distance between an effector to target cell was 86 μm (range 0–441 μm).

In addition, we evaluated whether any of the CAR/CXCR5$^+$ cells were also vRNA$^+$, given that the CD4$^+$ CAR/CXCR5 are presumably susceptible to infection by SIV. We found 1 of 689 CAR/CXCR5-T cells examined to be vRNA$^+$.

## Infusion of CAR/CXCR5-T cells into SIV-infected rhesus macaques is safe

We next investigated the safety and in vivo efficacy of CAR/CXCR5-T cell immunotherapy in SIV-infected ART-suppressed animals, compared to untreated control animals. The untreated

**DAPI CD20 CAR/CXCR5 T cells SIV RNA**

**Fig 2. CAR/CXCR5-T cells home to lymphoid follicles and recognize SIV-infected cells in vivo.** Location of CAR/CXCR5-T cells (red) within lymphoid tissues was determined in a chronically SIV-infected ART-naïve rhesus macaque, animal R14025 at 2 days post-treatment (DPT). Representative image of spleen tissue section showing duplex detection of CAR/CXCR5 construct (red) and SIV (pseudo-colored white) using RNAscope ISH combined with immunofluorescence using a custom-made probe for detection of CAR/CXCR5 construct and a probe specific for SIV. The white haze within the B cell follicle represents SIV virions trapped by the follicular dendritic cells (FDC) network. Scale bar = 100 µm. The right panels are enlargements showing an interaction between two CAR/CXCR5-transduced T cells and an SIV-infected cell. The tissue was stained with DAPI (blue), and anti-CD20 (green) to label B cells and delineate B cell follicles. Scale bar = 10 µm. Confocal images were collected using a 20× objective. The curves tool in Photoshop was used to increase the contrast of each image in a similar manner.

and treated animal groups included male and female animals, with age, weight, peak viral loads and CD4/CD8 frequencies, noted in Table 1. All animals were positive for the class I allele *Mamu-A1*001* and negative for protective alleles *Mamu-B*008* and *Mamu-B*017:01*.

CAR/CXCR5-T cells were produced using gammaretrovirus transduction of peripheral mononuclear cells (PBMCs). For the first treatment group (T1), CAR/CXCR5-T cells were generated using PBMCs collected from rhesus macaques during the chronic stage of infection. For the second treatment group (T2), CAR/CXCR5-T cells were generated from PBMCs collected prior to SIVmac251 infection. T1, T2, and control animals were suppressed with antiretroviral therapy (ART) that was initiated 63–68 days post-infection. Animals were released

**Table 1. Animal information.**

| Group | Animal | Age (yrs) | Sex | Weight (kg) | Peak VL Post-infection | Pre-ART CD4/CD8 frequency | ART treatment (days) | Necropsy (DPT) |
|---|---|---|---|---|---|---|---|---|
| T0 | R14025 | 3.6 | M | 5.5 | $3.80 \times 10^7$ | - | - | 2 |
| T1 | R01093 | 17 | F | 7.6 | $0.94 \times 10^7$ | 0.44 | 337 | 56 |
| T1 | Rh2526 | 9.8 | F | 5.0 | $4.74 \times 10^7$ | 0.76 | 379 | 70 |
| T1 | Rh2537 | 13 | F | 8.0 | $0.94 \times 10^7$ | 1.28 | 358 | 83 |
| C | R12049 | 6.3 | F | 6.4 | $6.37 \times 10^7$ | 1.06 | 337 | 56 |
| C | R11100 | 7.1 | F | 7.4 | $4.62 \times 10^7$ | 0.31 | 379 | 70 |
| C | R10002 | 9 | M | 11.2 | $2.75 \times 10^7$ | 0.77 | 358 | 83 |
| T2 | Rh2850 | 4.7 | M | 6.3 | $0.93 \times 10^7$ | 1.08 | 104 | 308 |
| T2 | Rh2853 | 7.1 | M | 9.2 | $2.00 \times 10^7$ | 1.29 | 76 | 77 |
| T2 | Rh2858 | 6.4 | M | 10.6 | $2.38 \times 10^7$ | 0.68 | 90 | 328 |

from ART at the time of CAR/CXCR5-T cell infusion and monitored for at least 60 days, as outlined in the study design shown in Fig 3. Blood and tissue samples were collected over time to monitor infused cells and SIV vRNA.

To evaluate the safety of the treatment, animals were monitored by veterinary staff twice daily for any signs of pain, illness, and stress by observing appetite, stool, behavior, and physical condition in response to the infused CAR/CXCR5-T cells. The animals exhibited no observable adverse effects after receiving the immunotherapeutic cells, and their weights were unaffected by the immunotherapeutic infusion. Necropsy reports noted no abnormalities in treated animals beyond those typical in SIV-infected animals.

A Luminex assay for monitoring cytokine levels after the cell infusion showed a transient spike in IL-6 and interferon gamma (IFN-γ) at 2 DPT in three of the six treated animals; and levels returned to normal by 6 DPT. None of the other treated animals or the control animals had cytokine levels above the limit of detection for the assay. (S2A–S2D Fig). Flow cytometry (S2E Fig) and quantitative polymerase chain reaction (qPCR) (S2F Fig) detected cells in

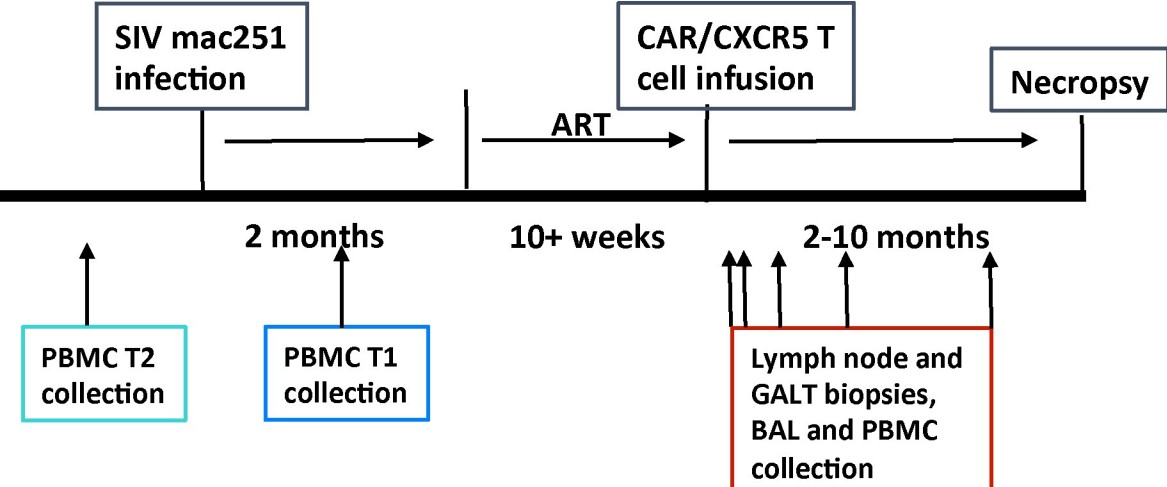

**Fig 3. A timeline of the Rhesus macaque pilot studies.** Animals were infected with SIV mac251 and ART suppressed. Cells were collected for transduction either post-infection (T1) or pre-infection (T2). ART was interrupted at the time of CAR/CXCR5 cell infusion. Blood and tissue samples were collected at indicated intervals post-infusion.

**Table 2. Cell infusions.**

| Group | Animal | Cells/kg infused | %CAR/CXCR5 | % CM | % CCR7$^+$ | % CD4$^+$ | % CD8$^+$ | % CD4$^+$CD8$^+$ |
|---|---|---|---|---|---|---|---|---|
| T0 | R14025 | $0.35 \times 10^8$ | 66.2 | 58.1 | 93.5 | 43.8 | 16.3 | 38.0 |
| T1 | R01093 | $0.8 \times 10^8$ | 55.0 | 50.3 | 66.5 | 22.3 | 46.5 | 28.8 |
| T1 | Rh2526 | $1.12 \times 10^8$ | 59.5 | 65.3 | 62.5 | 26.8 | 36.0 | 35.4 |
| T1 | Rh2537 | $0.75 \times 10^8$ | 57.4 | 66.9 | 59.0 | 42.3 | 27.3 | 28.9 |
| T2 | Rh2850 | $2.0 \times 10^8$ | 79.4 | 73.6 | 62.5 | 20.6 | 14.3 | 63.8 |
| T2 | Rh2853 | $1.06 \times 10^8$ | 68.6 | 54.5 | 44.6 | 32.3 | 15.3 | 48.3 |
| T2 | Rh2858 | $1.26 \times 10^8$ | 58.4 | 61.4 | 76.4 | 33.4 | 23.2 | 41.9 |

bronchoalveolar lavage (BAL) samples shortly after infusion; however, we detected no CTV-labeled cell accumulation in lung tissues (Fig 1). The cells likely accumulated in the BAL shortly after infusion due to pulmonary circulation. Upon necropsy, the lungs appeared healthy. The overall health of the animals and the transient nature of the cytokine spikes suggests that the infusion of autologous CAR/CXCR5-T cells is safe.

## Infused CAR/CXCR5-T cells were predominantly activated central memory T cells

T1 and T2 animals were infused with the T cell products at doses ranging from $0.8–2.0 \times 10^8$ cells/kg (Table 2). The infused cells were a mix of CD8 and CD4 T cells. Most of the infused cells expressed both the CAR and CXCR5 (range, 55–79.4%) and displayed primarily a central memory phenotype (range, 50.3–73.6%) (Table 2, S3 Fig). The majority of the central memory cells expressed C-C chemokine receptor 7 (CCR7) (range, 44.6–93.5%), a LN homing molecule [67–69].

## CAR/CXCR5-T cell infusion associated with reductions in viral loads

Viral loads were monitored after infusion of CAR/CXCR5-T cells into rhesus macaques. Untreated control animals (Fig 4A) showed a rapid rise in viral loads 10 to 14 days after ART cessation, followed by a slow decline over time that remained at detectable levels throughout the 56 to 83 day experiment. T1 animals (Fig 4B), in which cells were collected during chronic untreated infection, showed an initial immediate spike (noted with an asterisk in Fig 4B) in viral loads due to the presence of virus in the infused SIV-infected transduced cells (S4A Fig) and in the solution in which the cells were suspended for transport (S4B Fig) [70]. Viral loads dropped in all three treated animals—reaching undetectable levels in one of the three animals —and then began to rise. For the duration of the experiment, one of the three T1 treated animals maintained substantially lower viral loads compared to control untreated animals, and two of the three treated animals had undetectable viral loads at necropsy (56–83 DPT).

In T2 animals (Fig 4C), two of the three treated animals showed lower peak viral loads post-ART release compared to control untreated animals. One-month post-infusion (27–30 DPT), the viral loads in two of the three T1 and all three T2 animals were lower than untreated control animals, with the median viral load of T2 being nearly 2 logs below that of the untreated control animals (Fig 4D). An examination of total viral burden, represented by the area under the curve for days 0–56 post-treatment, also shows a nearly two log decrease in the T2 animals as compared to the control animals (Fig 4E). As an exception to the study design, which was planned to maintain animals for only 2–3 months post-cell infusion and ART release, we maintained two T2 animals for 10 months post-infusion to monitor long-term viral loads. During this time, the animals maintained control of infection, with viral loads oscillating

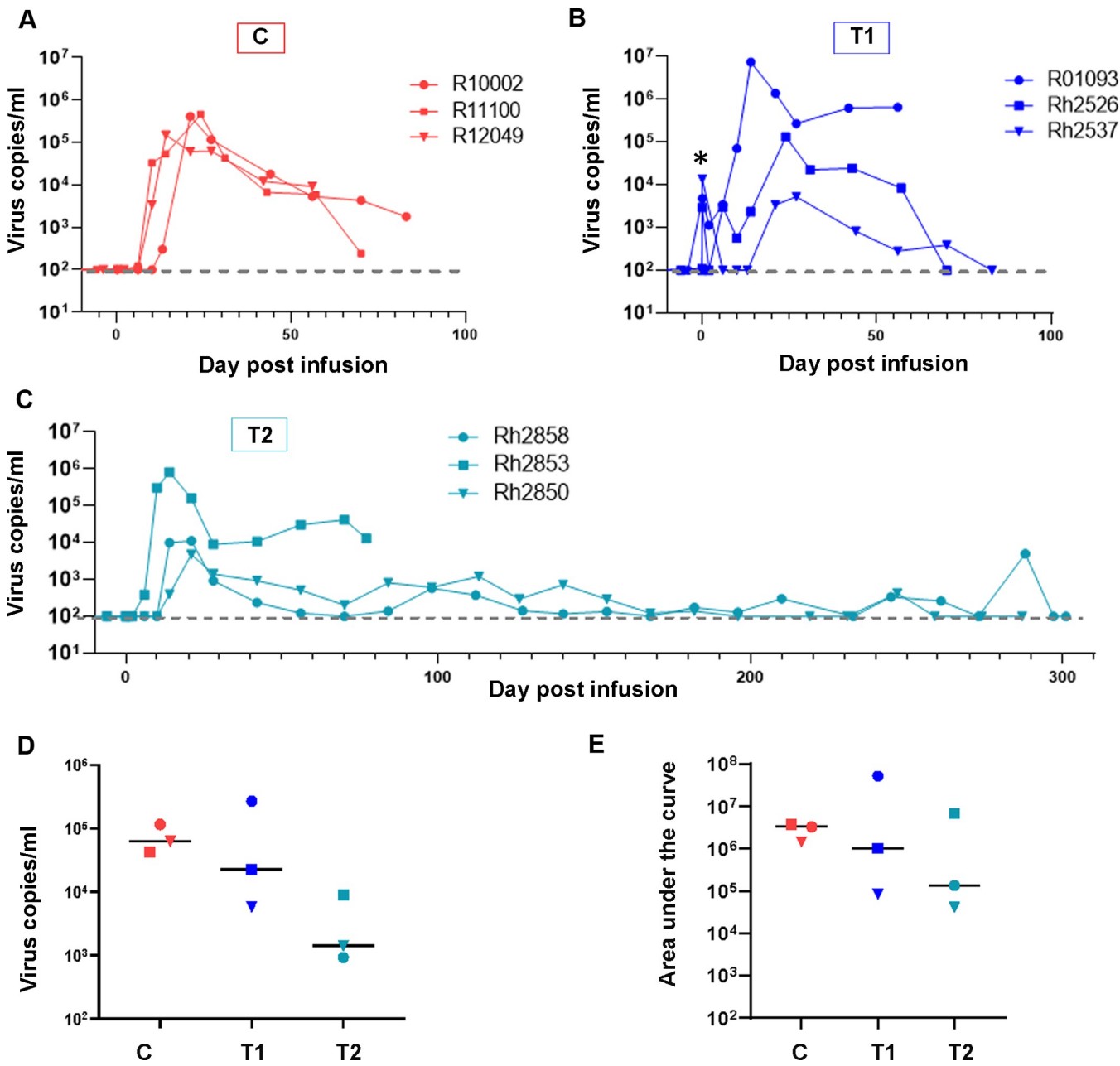

**Fig 4. Post-infusion viral loads.** Viral loads over time in (A) untreated control animals, (B) pilot study T1 animals and (C) pilot study T2 animals. Viral loads were determined by measurement of gag mRNA in plasma using reverse transcription (RT) polymerase chain reaction (PCR). The asterisk in panel B (T1) indicates the increase in viral load after infusion of the SIV containing T cell product. (D) Viral loads for the three animal groups one month post-infusion (27–30 DPI). The bar represents the median. (E) Total viral burden represented by area under the curve from day 0 to day 56 post-infusion. The bar represents the median for each of the groups.

between undetectable and very low levels (Fig 4C). We detected similar levels of naturally occurring immunodominant [71] SIV-specific (Mamu-A1*001/Gag-CM9) CD8$^+$ T cells in PBMCs of treated and untreated control animals at one-month post-infusion (S5 Fig). This finding suggests that the differences in viral loads between groups was not likely driven by differences in the endogenous immunodominant Gag/CM9 CTL response. Overall, these data

suggest that CAR/CXCR5-T cell therapy may be effective at reducing viral loads in SIV-infected rhesus macaques after ART cessation.

## CAR/CXCR5-T cells proliferate in vivo

At 2 DPT, CTV-labeled CAR/CXCR5-transduced cells showed evidence of proliferation in both the EF and F areas of LNs, showing doublets of cells and cells with decreased fluorescence intensity, indicating a loss of CTV with cell division. Cells within the F areas had an overall lower CTV fluorescence intensity than those in the EF (Fig 5A), suggesting that they had undergone further cell division. Similarly, at 2 DPT, RNAScope detection of CAR/CXCR5-T cells combined with immunofluorescence staining of LNs showed clusters of CAR/CXCR5-T cells at the edge of the follicles, suggestive of cell expansion (Fig 5B). These clusters were detected in over 50% of the follicles (range, 49–58%). In vivo proliferation of CAR/CXCR5-T cells was further confirmed at 6 DPT in LN sections in treated animals by a combination of RNAScope and Ki67 antibody staining to mark T cell activation and proliferation. We detected Ki67$^+$ CAR/CXCR5-T cells in both F and EF areas (Fig 5C). Levels of Ki67$^+$ CAR/CXCR5-T cells showed a median of 30% (range, 9–64%) of total CAR/CXCR5-T cells in F areas and a median of 36% (range, 13–44%) of total CAR/CXCR5-T cells in EF areas. Interestingly, the T2 animal demonstrating the greatest control (Rh2850), showed the highest percentage of F Ki67$^+$ CAR/CXCR5-T cells (64%) and the animal that lost control (Rh2853) showed the lowest percentage of follicular Ki67$^+$ CAR/CXCR5-T cells (9%) (Fig 5D).

## CAR/CXCR5-T cells localize to the follicle and persist for up to 28 days

We analyzed CAR/CXCR5-T cells in LN sections from T2 animals biopsied at 2, 6, 14, 28, and 60 DPT using RNAscope. There was a noticeable shift at 6 DPT to CAR/CXCR5-T cells primarily accumulating within B cell follicles (Fig 6B compared to Fig 5B). Quantification of CAR/CXCR5-T cells in the F and EF regions of LNs revealed that CAR/CXCR5-T cells were most abundant during the first week post-infusion, followed by a decline over time (Fig 6C). At 2 DPT, CAR/CXCR5-T cells were detected at similar levels in both F and EF areas, with a median of 28 cells/mm$^2$ (range, 26–30 cells/mm$^2$) in F areas and 30 cells/mm$^2$ (range, 23–37 cells/mm$^2$) in EF areas (Fig 6C). At 6 DPT the cells were detected predominantly in F areas with a median of 76 cells/mm$^2$ (range, 38–239 cells/mm$^2$) compared to a median of 4.3 cells/mm$^2$ (range, 3–34 cells/mm$^2$) in EF areas (Fig 6C) and resulted in relatively high effector to target ratios (E:T) of CAR/CXCR5 T cells: vRNA$^+$ cells in follicles, ranging from 38 to 149. Table 3 shows the E:T for each animal calculated by dividing the total CAR/CXCR5-T cells/mm$^2$ by the total vRNA+ cells/mm$^2$. The range of E:T for each individual F and EF area are also noted, as is the fold change difference of E:T in F compared to EF areas. Notably, the E:T ratios were substantially higher in F compared to EF in the two animals that went on to control infection, with a 5.4 fold increase in Rh2850, and 27 fold change for Rh2858. Whereas the treated animal that lost control showed similar E: T levels in F and EF areas. At 6 DPT, the median distance between effector CAR/CXCR5-T cells and target vRNA$^+$ cells in the follicle

**Table 3. T2 animals in vivo CAR/CXCR5$^+$: vRNA$^+$ E:T at 6 days post-infusion.**

| Animal # | Follicular | | Extrafollicular | | Fold change difference in F vs EF areas |
|---|---|---|---|---|---|
| | E:T | Range of E:T ratios | E:T | Range of E:T ratios | |
| Rh2850 | 38: 0 | 1–125 | 4.3: 0.61 | 0.9–13 | 5.4 |
| Rh2858 | 239: 1.6 | 41–279 | 3: 0.55 | 0.5–5 | 27 |
| Rh2853 | 76: 1.7 | 0.09–97 | 34: 0.55 | 2–47 | 0.7 |

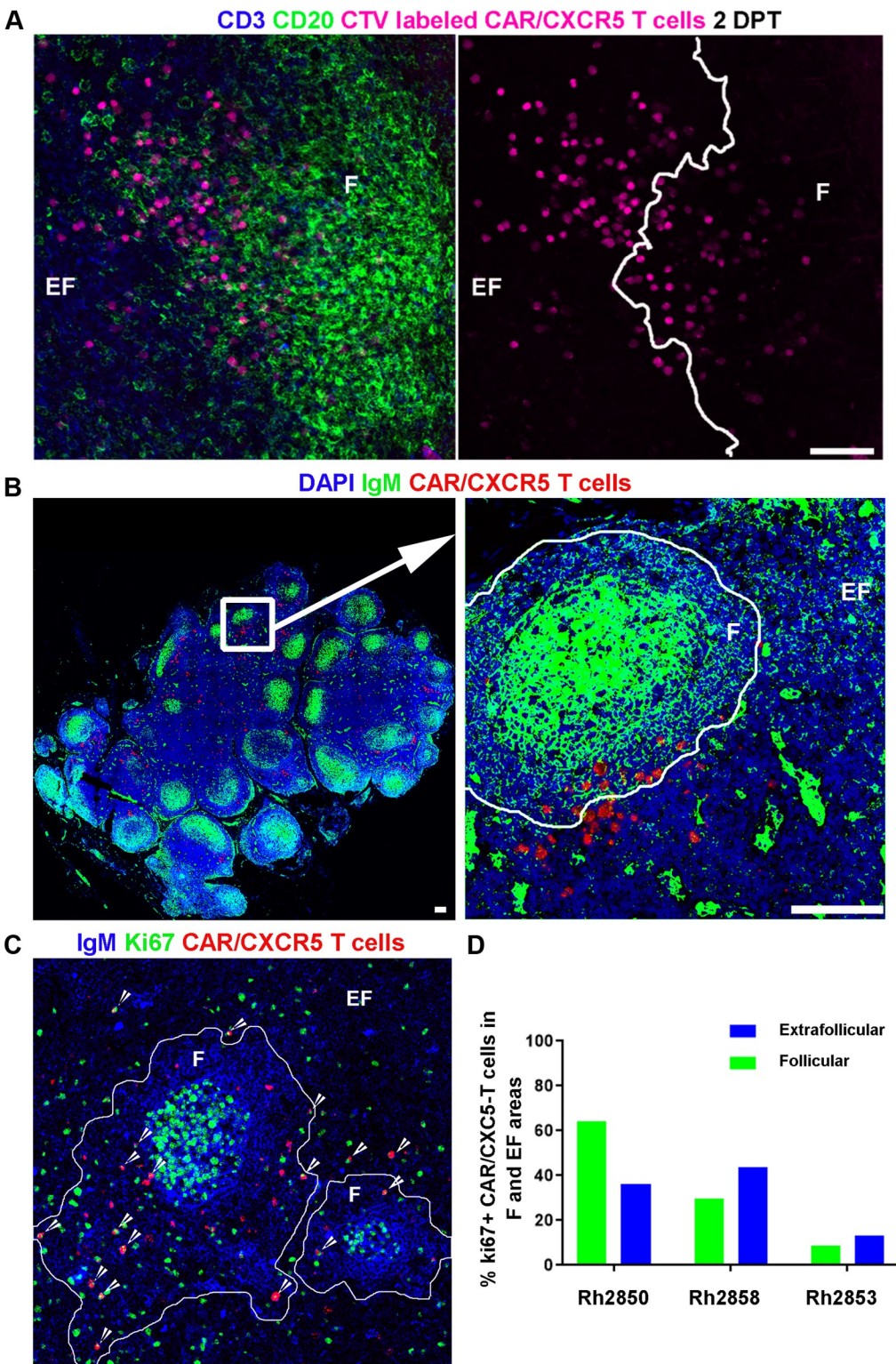

**Fig 5. CAR/CXCR5-T cells continue to expand in vivo.** CAR/CXCR5-T cells expanded at the edges of B cell follicles in vivo at 2 DPT and accumulated within B cell follicles at 6 DPT. (A) Representative image from LN tissue from Rh2850 stained 2 DPT showing CTV-labeled cell proliferation in the extrafollicular (EF) area near the follicular (F) zone. Tissues were stained with anti-CD3 (blue) to label T cells and anti-CD20 (green) to label B cells and delineate B cell follicles (F). The infused T cell product labeled with CTV is pseudo-colored magenta. Cells within F showed lower

fluorescence intensity indicating division. Scale bar = 50 μm. (B) Representative image of LN tissue section, from Rh2858 visualized using RNAScope ISH combined with IF, showing expansion of CAR/CXCR5-T cells at the edge of follicles at 2 DPT. The right panel is an enlargement from the left panel showing a cluster of CAR/CXCR5-T cells (red) that appear to be expanding at the edge of the follicle. Tissues were stained with DAPI (blue) and anti-IgM (green) to label B cells and delineate B cell follicles (F), with the more brightly stained germinal center in the center of the F. (C) Representative image of LN tissue from Rh2850, showing CAR/CXCR5-T cell (red) proliferation in F and EF at 6 DPT detected by RNAScope ISH combined with IF. Tissues were stained with anti-IgM (Blue) and anti-Ki67 (green) to mark activation and proliferation. B cell follicles are delineated with white lines. Scale bars = 100 μm. Confocal images were collected using a 20× objective. (D) Percentage of Ki67$^+$ CAR/CXCR5 T cells in the F (green) and EF (blue) areas for each of the T2 animals.

was 60 μm with a range of 0–170 μm. At 14 DPT, the F:EF ratio of CAR/CXCR5-T cells in lymph nodes increased; however, the overall frequency of cells sharply declined in all treated animals, with a median of 2.3 cells/mm$^2$ (range, 0.32–7.6 cells/mm$^2$) in F and 0.4 cells/mm$^2$ (range, 0–0.57 cells/mm$^2$) in EF areas (Fig 6C). By 28 DPT, cells were only detected in F areas of one animal (Rh2850; 1.17 cells/mm$^2$), with no cells detected at 60 DPT in any of the examined sections of the treated animals (Fig 6C). Notably, the animal that lost viral control (Rh2853) showed the fastest and steepest decline in levels of CAR/CXCR5-T cells over time relative to two animals that controlled infection.

Most follicles in the LNs had detectable CAR/CXCR5-T cells during the first week post-treatment. In fact, at 6 DPT; a median of 96% (range, 90–100%) of follicles examined had CAR/CXCR5-T cells (Fig 6D). These levels declined in all animals at subsequent timepoints.

Similar RNAScope analyses were carried out with samples from the T1 animals to determine the location, abundance, and persistence of CAR/CXCR5-T cells within lymph node tissue. However, samples were limited from these animals. As was seen with the T2 animals, the T1 animals contained higher levels of CAR/CXCR5-T cells in the F areas than the EF areas (S6A Fig) and the CAR/CXCR5-T cells were detected up to 28 DPT (S6B Fig). However, the levels of CAR/CXCR5-T cells were lower than those detected in T2 animals.

Because transduced cells for T1 were derived from PBMCs collected during the chronic phase of infection, we performed in situ tetramer staining on lymph node biopsies, and flow cytometry on PBMCs collected after infusion to determine whether any of the CAR/CXCR5-T cells expressed the immunodominant Mamu A01 Gag/CM9 responsive TCR. We found few to no Gag/CM9$^+$/CTV$^+$ cells in the lymph nodes indicating that very few if any of the infused T cells were immunodominant Gag/CM9$^+$ T cells. Furthermore, flow cytometry of the CAR/CXCR5-T cells in blood collected post-infusion revealed that none of the CAR cells detected were Gag/CM9$^+$.

Examination of PBMCs using both flow cytometry and qPCR revealed a similar pattern of CAR/CXCR5-T cell persistence. Flow cytometric analysis detected CAR/CXCR5-T cells in isolated PBMCs up to 14–21 DPT (Figs 6E and S3). Similar examination of control animal PBMC and BAL samples did not detect CD4$^+$MBL$^+$ cells (Figs 6E and S3), verifying flow cytometry as a robust method to determine CAR/CXCR5-T cell frequency in the PBMC and BAL. Genomic DNA PCR detection of CAR/CXCR5-T cells in PBMCs showed a similar decline in cell number by day 14 (Fig 6F). While no control animal samples were available for analysis by PCR, genomic DNA from non-transduced cell samples were used in the assay and no specific product was detected. In addition, we found a strong positive correlation between levels of follicular CAR/CXCR5-T cells in LN tissue in situ and the frequency of CD4-MBL$^+$ CAR cells detected in PBMCs by flow cytometry (S7 Fig). This finding suggests that the cells have similar persistence in peripheral blood and tissue, and that detection of CAR/CXCR5-T cells in peripheral blood may be a useful surrogate marker for levels and persistence of CAR/CXCR5-T cells in lymphoid tissues.

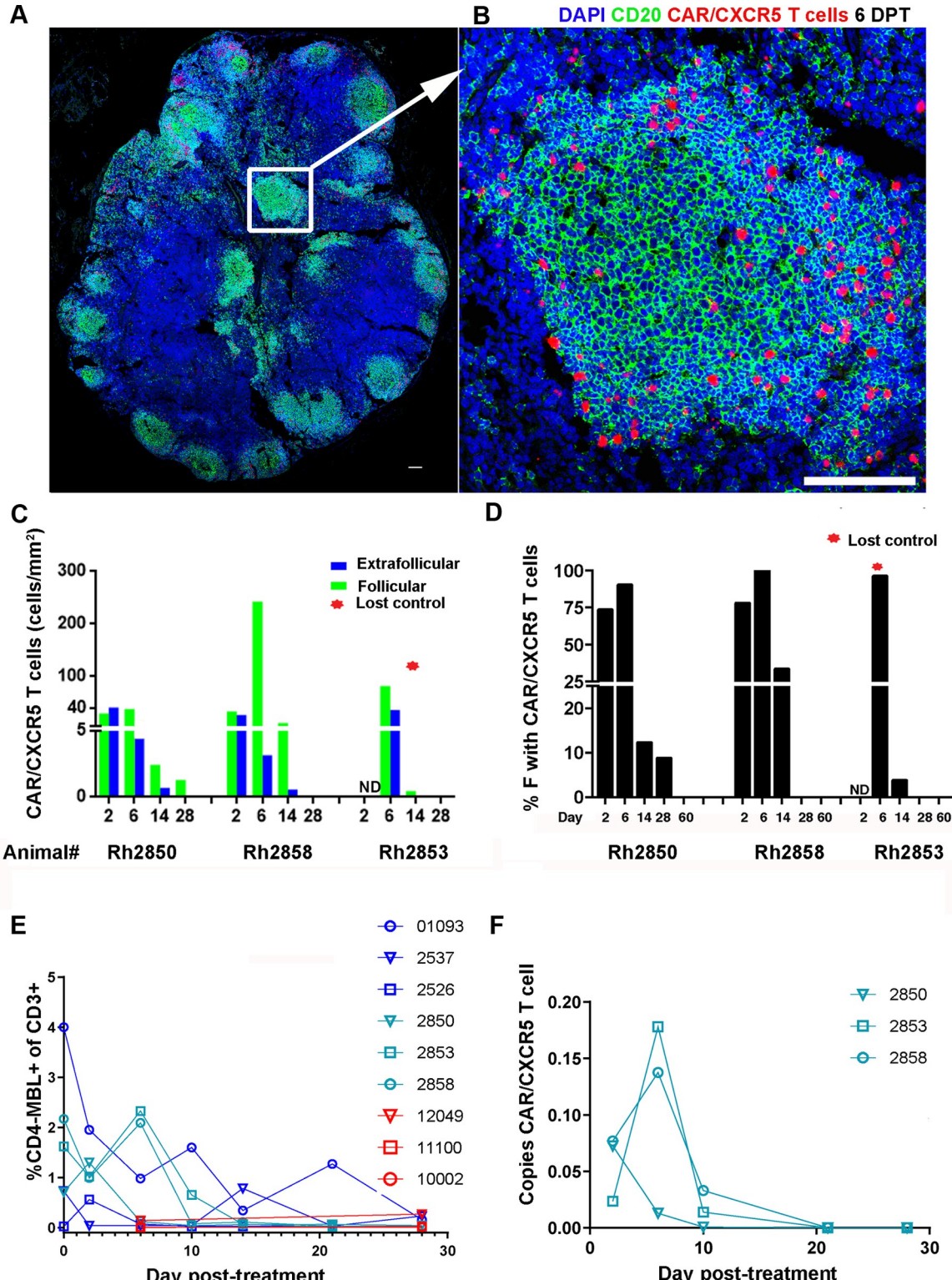

**Fig 6. CAR/CXCR5-T cells localize to over 90% of the follicles and persist for up to 28 days.** CAR/CXCR5-T cells successfully homed to over 90% of the B cell follicles at 6 DPT and persisted for up to 28 DPT in SIV-infected ART-suppressed/released animals. Representative images of LN tissue section from Rh2858, showing CAR/CXCR5-T cells (red) detected using RNAScope ISH combined with IF using a custom-made probe for detection of the CAR/CXCR5 construct. (A) Confocal image showing a whole LN tissue section. (B) Enlargement of the delineated area in (6A) showing that CAR/CXCR5-transduced T cells (red) successfully homed to a B cell follicle

(green). The tissue was stained with DAPI (blue), and anti-CD20 (green) to label B cells and delineate B cell follicles. The confocal image is collected with a 20× objective. Scale bar = 100 μm. (C) Levels of CAR/CXCR5-T cells over time after infusion in F (green) and EF areas (blue) of LN. The animal that lost control of the virus infection is marked with a red star. Samples not determined are marked ND. (D) Percentage of follicles that contained CAR/CXCR5-T cells over time post-infusion. (E) Frequency of CD4-MBL+ cells in CD3+ PBMCs as determined by flow cytometry and (F) copies of CAR in the total cell population in PBMCs as determined by quantitative real-time PCR at the indicated time points post-infusion.

## In vivo levels of viral RNA appear to be impacted by CAR/CXCR5-T cells

We determined the levels of vRNA in the three T2 animals and three control animals at 28 DPT (Fig 7). The two treated animals that exhibited sustained control of SIV infection (Rh2850, Rh2858) showed few to no SIV vRNA+ cells at 28 DPT in F and EF areas compared to abundant SIV vRNA+ cells in untreated control animals and the T2 animal that did not control the infection (Rh2853) (Fig 7A and 7B). Notably, we detected no CAR/CXCR5-T cells that were SIV vRNA+ in the examined sections. In addition, Rh2850 and Rh2858 animals had lower percentages of follicles with free virions trapped by the FDC network than untreated control animals, or the treated animal that lost control (Rh2853) (Fig 7C). In fact, Rh2850 had no detectible FDC associated virions in any follicles, and only one of 30 follicles showed FDC-associated virions in Rh2858, whereas most follicles showed FDC trapped virions in untreated control animals and the treated animal that lost control. These findings suggest that the immunotherapeutic cells may have led to sustained reductions in vRNA in the treated animals.

## Discussion

HIV and SIV viral replication is concentrated in lymphoid B cell follicles [24–30] during chronic infection, with infected Tfh in follicles representing a major barrier to HIV eradication [24–27,31,32,63]. The failure of virus-specific CD8+ T cells to accumulate in large numbers within the B cell follicles of HIV-infected individuals and SIV-infected rhesus macaques appears to be a major mechanism allowing for persistent follicular viral replication [26,27,32,42,43,72]. In addition, increasing levels of SIV- and HIV-specific T cells within B cell follicles are associated with viral control [27,46,73,74]. These findings led us to hypothesize that infusion of T cells engineered to co-express a potent SIV-specific CAR along with the B cell homing molecule, CXCR5, will control SIV-infection by reducing viral replication in follicles. We tested this hypothesis in an SIV-infected rhesus macaque model of HIV infection, in which SIVmac251-infected animals were ART suppressed prior to treatment, and ART removed on the day of treatment with CAR/CXCR5-T cells. Combining a potent SIV-specific CAR with the B cell follicle homing properties of CXCR5 on T cells can overcome the limitation of the endogenous virus-specific CD8+ T cell immune response to SIV and lead to control of viremia after ART interruption.

This study provided preliminary evidence of both the efficacy and the safety of autologous CAR/CXCR5-T cell immunotherapy. CAR/CXCR5-T cells successfully homed to B cell follicles and interacted with SIV-infected cells in vivo. The cells proliferated in situ and accumulated primarily in F areas. Five of six treated animals showed lower viral loads at one-month post-infusion and had fewer follicular vRNA+ cells in LNs compared to untreated animals. Apart from a transient increase in inflammatory cytokines, none of the animals infused with CAR/CXCR5-immunotherapeutic T cells had an adverse reaction to the infusion. Taken together, these pilot studies in the rhesus macaque model of HIV infection suggest that autologous CAR/CXCR5-T cell therapy may be a viable tool for the treatment of HIV infection in ART-suppressed individuals after ART cessation.

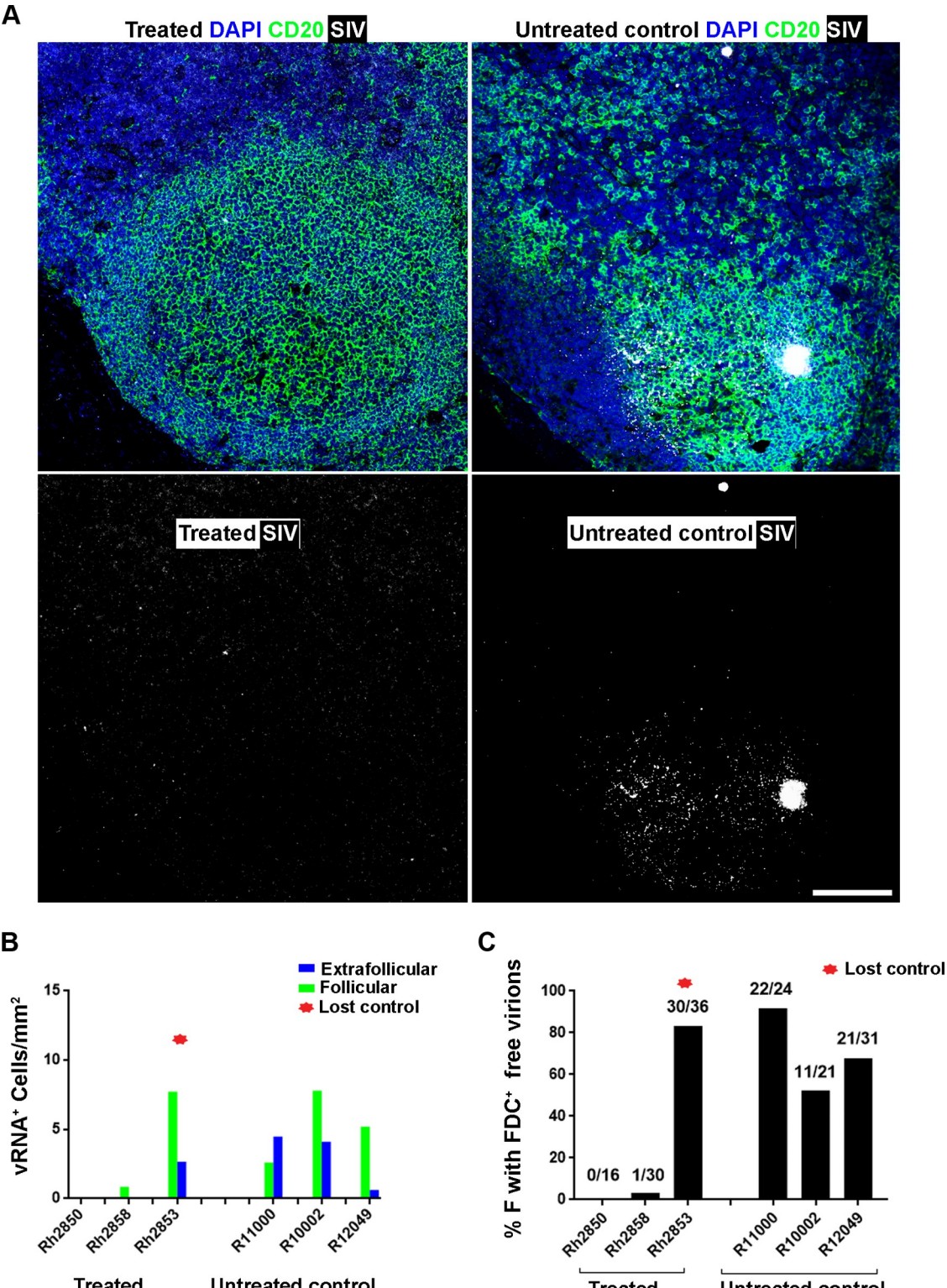

**Fig 7. Infusion of CAR/CXCR5-T cells into SIV-infected rhesus macaques results in lower viral loads post-ART interruption.**
At 28 DPT, treated animals showed a reduction in viral (v)RNA compared to untreated control animals. (A) A representative image from a LN tissue section showing the abundance of SIV vRNA⁺ cells and free virions trapped by follicular dendritic cells network (pseudo-colored white) detected using RNAscope ISH combined with IF in treated from Rh2850 (left panels) versus untreated control from R11002 (right panels). The tissue was stained with DAPI (blue), and anti-CD20 (green) to label B cells and delineate B

cell follicles. The white haze within the B cell follicle represents SIV virions trapped by the follicular dendritic cells (FDC) network. Confocal images were collected with a 20× objective. Scale bar = 100 μm. (B) Levels of viral RNA in F and EF areas. The animal that lost viral control of the virus infection is marked with a red star. (C) The percentage of follicles with free virions bound by FDC.

This study used cells collected either during chronic infection (T1) or collected prior to infection (T2) to produce CAR/CXCR5-T cells. We did not use cells from ART-suppressed animals because we and others observed reductions in the transduction efficiency of T cells from ART-treated subjects [70,75]. Such decreased transduction efficiency may be due to interference with reverse transcription and integration of the gammaretroviral vector by the residual ART drugs. Expression of the transgenes in T cells from ART-suppressed subjects may be improved with alterations in the transduction protocol, use of vectors for virus production that are not impacted by ART drugs or through the use of alternative methods of gene transfer [70].

Our previous studies examining the function of (CD4-MBL) CAR/CXCR5-T cells indicated that the CXCR5 molecule facilitated migration to B cell follicles [61]. We demonstrated, in an ex vivo tissue migration assay, that CAR/CXCR5 expressing cells preferentially accumulated in lymphoid follicles while cells expressing CAR alone did not. Ayala et al. previously showed that engineered CXCR5 expression on T cells induced cell migration into lymphoid follicles in vivo in rhesus macaques, whereas cells not expressing CXCR5 accumulated in extrafollicular T cell zones [60]. This study extends those findings, demonstrating that the addition of an antiviral CAR on CXCR5-transduced T cells leads to similar migration and accumulation within B cell follicles.

Shortly after infusion, CAR/CXCR5-T cells proliferated in vivo. At 2 DPT, we detected clusters of replicating CAR/CXCR5-T cells, often located at the edge of the follicles, in 50% of the examined follicles. The CAR/CXCR5-T cells appeared to replicate initially next to follicles, then enter into the follicles as suggested by progressively decreasing CTV staining from EF to F areas. Ki67 staining confirmed the in vivo proliferation of CAR/CXCR5-T cells and showed that the cells were replicating in both EF and F areas. It is unclear what cues instigated the in vivo proliferation. Factors that may have contributed include: the culturing conditions prior to infusion, the detection of antigen in vivo by CAR/CXCR5-T cells, or other proliferation cues.

We have previously shown that CAR/CXCR5-T cells readily target and suppress SIV-infected T cells in vitro [61]. This in vivo study showed detection of (CD4-MBL) CAR/CXCR5-T cells directly interacting with SIV-infected cells within lymphoid follicles. These interactions provide evidence that CAR/CXCR5-T cells specifically targeted SIV vRNA$^+$ cells, and likely led to the death of these infected cells. We also found that the distance of vRNA + cells to the nearest CAR/CXCR5-T cell ranged from 0–441μm in the sections analyzed. These tissue sections captured a snapshot of the location of the cells at the time of biopsy. Importantly, T cells move an average speed of 10 μm per minute in secondary lymphoid tissues [76]. Thus, in theory, the CAR/CXCR5-T cells were close enough to each of the SIV vRNA$^+$ cells to track down and kill them within a relatively short period of time.

Since the CAR does not require antigen presented in the context of an MHC molecule, it is possible that CAR/CXCR5-T cells may not only attack HIV/SIV producing T cells in vivo but may also act to clear the FDCs containing bound SIV. However, it was recently reported that CD4-MBL CAR-T cells did not target FDCs bearing HIV bound immune complexes in vitro [77], suggesting that the CAR-T cells may not target FDC bearing virus in vivo. We noted little or no virus particles trapped by FDC in the LNs of the T2 animals that went on to show long-term control of the virus and cannot rule out the possibility that the CAR/CXCR5-T cells were able to recognize and remove FDC bearing virions. Alternatively, and perhaps more plausibly,

the killing of SIV-producing cells in follicles may have led to a reduction in the seeding of FDCs with virions, explaining the absence of SIV loaded FDC.

Historically, engineering CD8[+] T cells to express CD4 presents a challenge for HIV treatment due to the likelihood that enhanced CD4 expression would make the transduced CD8[+] T cells susceptible to HIV infection [12,78–81] owing to abundant expression of the CCR5 coreceptor [82]. To overcome this issue, we used a bispecific CAR construct, (CD4-MBL) CAR/CXCR5 [12,23]. The MBL moiety of the CAR creates a steric hindrance that prevents the CD4 of the CAR from acting as an entry receptor [23]. Nonetheless, CD4[+] T cells transduced with CAR and CXCR5 are presumably susceptible to SIV/HIV infections. Interestingly, we only detected 1 SIV vRNA[+] CAR/CXCR5[+] cell out of 689 CAR/CXCR5[+] cells examined in the treated chronically infected animal. These data suggest that either the CD4[+] CAR/CXCR5-T cells were not readily infected or, if infected, were rapidly cleared.

Infusion of CAR/CXCR5-T cells appeared to cause no ill effects in the animals, as indicated by veterinary health records and by unremarkable necropsy reports including a lack of inflammation in the lung. Three of six treated animals showed an increase in serum IL-6 and IFN-γ at 2 DPT; however, the effect was transient, did not manifest as detectable illness, and may have actually been a measurement of early in vivo activity of the CAR-T cells [83–85]. The animals exhibited no neurotoxic symptoms or signs of fever and weight loss, which were described in a recent study of neurotoxicity following CAR-T cell infusion [86]. Previous rhesus macaque adoptive transfer studies found that a significant fraction of the infused T cells localize to the lung with limited to undetectable persistence in blood or lymphoid tissues [87,88]. In lung tissues, we detected very low levels of CTV-CAR/CXCR5-T cells in vivo from the animal sacrificed at 2 DPT. We detected CAR/CXCR5-T cells in BAL fluid in four of the six treated animals between 2 and 14 DPT; however, this accumulation was transient as no cells were detected at 28 DPT.

We have previously shown that CAR/CXCR5-T cells produced from chronically infected animals, while effective at suppressing SIV in culture, retain residual virus [70]. T1 animals received CAR/CXCR5-T cells generated using PBMCs from chronically SIV-infected animals. Thus, it was not surprising that T1 animals showed a spike in viral load immediately post-infusion. Nonetheless, even with the infusion of virus along with the CAR/CXCR5-T cells, one of the three treated animals maintained viral load below $10^4$ copies/mL throughout the study, and two had undetectable viral loads at necropsy.

While our animal numbers per group were too small to perform statistical analyses between groups, we found that two of the three T2 animals maintained viral loads below $10^4$ copies/mL for the entire study, with levels below or near the level of detection for up to 10 months post-infusion. We detected very little to no SIV RNA[+] cells in both F and EF areas of LNs from these two animals one-month post-treatment. We cannot be certain that this remission resulted from infusion of the CAR/CXCR5-T cells. However, while not unheard of, it is rare for SIVmac239 or SIVmac251 infected *Mamu-A1*001* rhesus macaques to spontaneously control viral replication for 10 months after interruption of ART [89–91]. Since the CAR/CXCR5-T cells did not persist beyond one month, it is unclear how the two treated animals maintained control for the subsequent months. The CAR/CXCR5-T cells may have provided early control and clearance of the recrudescent SIV producing cells post-ART release, allowing the endogenous immune response to be effective in maintaining low or undetectable viral loads in these animals. The CAR/CXCR5-T cells may have also effectively cleared emerging virus producing cells in follicles leading to decreased deposition of virus on FDCs and decreased viral production over time.

The T2 animal that did not control viral infection showed several important differences from the two controller animals. In the non-controlling animal, the infused cells showed

decreased expansion in vivo, declined more quickly and were more abundant in the BAL. This animal also had the lowest fold change difference in F vs. EF in vivo E:T ratios of CAR/CXCR5 T cells: vRNA⁺ cells. Additionally, the non-controlling animal had a relatively lower percentage of central memory cells and lower CCR7 expression than the animals that controlled plasma viremia post-infusion. It is possible that, due to differences in central memory phenotype, the cells did not persist as long, or were not present at sufficient levels, at the time of viral recrudescence, and failed to exert an effect on viral load.

CAR/CXCR5-T cells accumulated predominantly in lymphoid tissues, where they reached peak accumulation at 6 DPT and persisted for up to 28 days. The preferential homing of infused CAR/CXCR5-T cells to lymphoid tissues in our study may be due to relatively high frequency of central memory cells (median 63%) [68]. Overall, no CAR/CXCR5-T cells were found in the blood or tissues past 28 DPT. Many adoptive transfer studies utilize a lymphodepletion regimen using cytotoxic agents, such as Cytoxan, to create room for the adoptively transferred cells to implant [92–95], and such a conditioning regimen in future studies may allow greater expansion and persistence of CAR/CXCR5-T cells. Alternatively, persistence might be improved by the addition of antigen-expressing cells at, or shortly after, the time of CAR-T cell infusion to stimulate CAR-T cells in vivo, as was recently reported by Rust et al [96].

To achieve better long-term control of infection, CAR/CXCR5-T cell immunotherapy could also be combined with other strategies aimed at eliminating the HIV/SIV reservoir. Such therapies might include latency reversal agents [62,97] or agents that modify the immune system [98], such as an IL-15 superagonist [99].

These studies provide preliminary evidence of the safety and potential efficacy of CAR/CXCR5-T cells in a rhesus macaque model of HIV infection. Future studies with larger numbers of animals are essential to definitively determine the safety and efficacy of this intervention. With their ability to localize to the site of viral replication and to interact with virally infected cells, these immunotherapeutic cells have the potential to play an important role in the long-term cure of HIV infection without the use of life-long ART.

## Materials and methods

### Ethics statement

Rhesus macaques were housed at the Wisconsin National Primate Research Center (WNPRC). All procedures were approved by the University of Wisconsin-Madison College of Letters and Sciences and Vice Chancellor for Research and Graduate Education Centers Institutional Animal Care and Use Committee (IACUC protocol number G005529). The animal facilities of the Wisconsin National Primate Research Center are licensed by the US Department of Agriculture and accredited by AAALAC. Animals were monitored twice daily by veterinarians for any signs of disease, injury, or psychological abnormalities. At the conclusion of the study, animals were humanely euthanized by anesthetizing with ketamine (at least 15 mg/kg IM) or other form of WNPRC veterinary approved general anesthesia followed by an IV overdose (at least 50 mg/kg or to effect) of sodium pentobarbital or equivalent as approved by a WNPRC veterinarian. Death was defined by stoppage of the heart as determined by a qualified and experienced person using a stethoscope to monitor heart sounds from the chest area, as well as all other vital signs, which can be monitored by observation.

### Animal study design

The studies used 10 rhesus macaques that were positive for the class I allele *Mamu-A1*001* but negative for *Mamu-B*008* and *Mamu-B*017:01*. The initial treated animal was chronically

infected with SIVmac239 for 20 months prior to treatment. The animal was necropsied at day 2 post-infusion to determine the abundance and localization of the infused cells. Pilot study 1 and 2 treated animals (T1 and T2) and control untreated animals (C) (n = 3 per group) were infected intrarectally with SIVmac251 ($1 \times 10^8$ viral RNA). ART consisting of 5.1 mg/kg Tenofovir Disoproxil Fumarate (TDF) (Gilead), 40 mg/kg Emtricitabine (FTC) (Gilead) and 2.5 mg/kg Dolutegravir (DTG) (Viiv) was formulated at Beth Israel Deaconess Medical Center (BIDMC). ART was initiated at day 63–68 post-infection and continued daily until the day of cell infusion. Blood samples were drawn biweekly to monitor viral loads, and all animals had undetectable viral loads at the time of infusion. Animals were ART-suppressed for times indicated in Table 1. The animals did not undergo a lymphodepletion regimen prior to infusion of CAR/CXCR5-T cells. PBMCs were collected by density gradient centrifugation from blood draws either post-infection for T1 or pre-infection for T2. PBMCs were cryopreserved in CryoStor CS5 (BioLife Solutions Inc.) at a concentration between 4 and 20 million cells/mL and transported and stored in liquid nitrogen until use.

## Cell manufacturing and infusion

The CD4-MBL CAR/ CXCR5 construct was described previously [61]. The bi-specific CAR contains rhesus codon-optimized CD4 and MBL domains, which leads to specificity for SIV, linked to extracellular hinge, transmembrane and co-stimulatory domains of rhesus CD28 and the activation domain of rhesus CD3 zeta [12,23]. The follicular homing receptor, CXCR5, is linked to the CAR with a self-cleaving peptide, P2A. The genes were subcloned into the pMSGV1 gammaretrovial vector. Gammaretroviruses were produced by lipofectamine-mediated transfection of 293T cells by cotransfection with pBS-CMV-gagpol (a gift from Dr. Patrick Salmon, Addgene plasmid #35614) [100], a plasmid encoding RD114 [101], and pMD.G encoding VSV-G (a gift from Dr. Scott McIvor) [102] at ratios of 3:1:1:0.4, respectively [61]. CD4-MBL CAR/CXCR5-T cells were manufactured using the CD4-MBL CAR/CXCR5 gammaretrovirus as outlined previously [70,103]. Briefly, the PBMC were thawed and stimulated with plate-bound anti-CD3 (clone FN18) and soluble anti-CD28 for two days prior to retronectin-mediated transduction with gammaretroviral vector at an MOI of 0.5. Two days after transduction, the cells were placed in G-Rex 6 well plates (Wilson Wolf Corporation) and expanded for 4 days. All media contained 50 U/ml IL-2. Prior to infusion, cells were collected, washed, stained with CTV, an intracellular fluorescent dye, resuspended at a density of $2 \times 10^7$ cells/mL in RPMI for T0 and in PBS containing 10% autologous serum for all other animals, packed on ice and transported to the WNPRC. The T cell products were infused intravenously over 20 min while the animals were sedated. A veterinarian was present during the entire infusion. The dose of cells ranged from 0.35 to $2 \times 10^8$ cells/kg (Table 2). Following infusion, animals were evaluated for signs of pain, illness, and stress observing appetite, stool, typical behavior, and physical condition by the staff of the Animal Services Unit at least twice daily. The weight of the animals was monitored routinely throughout the protocol.

## Tissue, blood, and cell collection

Blood samples were drawn for viral load determination immediately before and after infusion and on days 2, 6, 10, 14 and then biweekly until necropsy. Complete blood counts (CBC) were monitored biweekly throughout the experiment. LN biopsies and BAL samples were collected on days 2, 6, 14, 28 and 60–69 post-infusion. Colon and rectal biopsies were collected on days 2, 14, 28 and 60–69 post-infusion. Animals were necropsied between day 56 and 328 post-infusion.

## Viral load determination

Viral loads were measured by Virology Services (WNPRC). vRNA was isolated from plasma samples using the Maxwell Viral Total Nucleic Acid Purification kit on the Maxwell 48RSC instrument (Promega, Madison WI). vRNA was then quantified using a highly sensitive qRT-PCR assay based on the one developed by Cline et al. [104]. RNA was reverse transcribed and amplified using TaqMan Fast Virus 1-Step Master Mix qRT-PCR Master Mix (Invitrogen) on the LightCycler 480 or LC96 instrument (Roche, Indianapolis, IN) and quantified by interpolation onto a standard curve made up of serial ten-fold dilutions of in vitro transcribed RNA. RNA for this standard curve was transcribed from the p239gag_Lifson plasmid, kindly provided by Dr. Jeffrey Lifson, (NCI/Leidos). The final reaction mixtures contained 150 ng random primers (Promega, Madison, WI), 600 nM each primer and 100 nM probe. Primer and probe sequences are as follows: forward primer: 5′- GTCTGC GTCATCTGGTGCATTC-3′, reverse primer: 5′-CACTAGCTGTCTCTGCACTATG TGTTTTG-3′ and probe: 5′-6-carboxyfluorescein-CTTCCTCAGTGTGTTTCACTTTCT CTTCTGCG-BHQ1-3′. The reactions cycled with the following conditions: 50˚C for 5 min, 95˚C for 20 s followed by 50 cycles of 95˚C for 15 s and 62˚C for 1 min. The limit of detection of this assay is 100 copies/mL.

## Flow cytometry

Multiparametric flow cytometry was performed on fresh, transduced PBMCs and on thawed PBMCs or BAL cells collected post-infusion with monoclonal antibodies cross-reactive in rhesus macaques to detect (CD4-MBL)CAR/CXCR5-T cells and SIV-specific T cells. Cells were incubated with Live/Dead NIR (Invitrogen); Alexa Fluor 700 mouse anti-human CD3 (SP34-2), FITC, Brilliant Violet 650 mouse anti-human CD4 (M-T477), Brilliant Violet 510 mouse anti-human CD8 (RPA-T8), PerCP/Cy5.5 mouse anti-human CD95 (DX2) and Brilliant Violet 605 mouse anti-human CD28 (28.2) (BD Biosciences); Phycoerythrin (PE) mouse anti-human CXCR5 (MU5UBEE) (eBiosciences); MBL (3E7) (Invitrogen) conjugated to Alexa Fluor 647. To detect SIV-specific CD8$^+$ T cells, samples were incubated with PE-labeled GAG-CM9 (NIH Tetramer Core) at 37˚C for 15 min.

## CAR/CXCR5-T cell PCR

DNA qPCR was used to determine the quantity of CAR-T cells in PBMCs and BAL. Genomic DNA was isolated from freshly thawed PBMCs or BAL collected post-infusion using the DNeasy Blood and Tissue kit (Qiagen). PCR primers were designed to specifically bind to the junction of the CD4 and MBL fragments of the CAR to avoid recognition of endogenous CD4 or MBL. Primers were checked for specificity in rhesus macaque and Homo sapiens for Refseq RNA, Refseq mRNA, and Refseq representative genomes in NCBI primer blast. The assay included 300 nM concentrations of the following primers:

CAR forward primer 5′-ATATTGTGGTCCTGGCCTTTCA-3′; CAR reverse primer 5′-AAGAATTTGTTTCCGACCTGCC-3′; albumin forward primer 5′-TGCATGAGAAAACG CCAGTAA-3′; albumin reverse primer 5′-ATGGTCGCCTGTTCACCAA-3′. PCR was run on a CFX96 thermal cycler (BioRad) with a program of one cycle of denaturation at 95˚C for 2 min, followed by 40 cycles of 95˚C for 10 s and 60˚C for 30 s. No specific product was detected in non-transduced cells. An amplified DNA fragment of the CAR was used in a standard curve to determine the copy number of the CAR. Albumin, which is present as two copies per cell, was used to determine cell number. The limit of detection was 2 copies of CAR DNA per $10^5$ cells.

## Luminex assay

Serum samples were stored at -80°C prior to analysis. Samples were tested by the Cytokine Reference Laboratory (University of Minnesota) using the magnetic bead set PRCYTOMAG-40K (EMD Millipore). Samples were analyzed for Non-Human Primate (NHP)-specific TNFα, IFNγ, IL-6 & IL-2 using the Luminex platform and performed as a multi-plex. Fluorescent color-coded beads coated with a specific capture antibody were added to each sample. After incubation and washing, biotinylated detection antibody was added followed by phycoerythrin-conjugated streptavidin. The beads were read on a Luminex instrument (Bioplex 200). Samples were run in duplicate and values were interpolated from five-parameter fitted standard curves.

## Singleplex RNAScope in situ hybridization and immunohistochemistry

RNAScope in situ hybridization utilized the 2.5 HD Reagent RED kit (Advanced Cell Diagnostics) as described previously [65,66,105] with modifications. FFPE tissue sections (5 μm) on slides were deparaffinized by baking 1 h at 60°C, rinsing in xylene followed by absolute ethanol and air-drying. Sections were boiled in RNAScope 1× Target Retrieval buffer (Advanced Cell Diagnostics) for epitope retrieval. Sections were then washed in dH$_2$O, dipped in absolute ethanol and air-dried. Following a protease pretreatment, sections were rinsed with dH$_2$O and hybridized overnight at 40°C with one of the following probes (all from Advanced Cell Diagnostics): SIVmac239 no-env antisense probe or a custom-made probe for the gammaretroviral vector to detect the CAR/CXCR5-transduced cells, DapB probe as a negative control probe or Macaca mulatta peptidylprolyl isomerase B (cyclophilin B) probe served as a positive control. Sections were then washed with 0.5× RNAScope wash buffer (Advanced Cell Diagnostics) and incubated with amplification reagents (1–6) according to the manufacturer's instructions. For chromogenic detection, sections were incubated with 120 μL of fast Red chromagen solution and washed as recommended by the manufacturer. For immunofluorescence staining, sections were blocked with 4% normal goat serum (NGS) and incubated overnight with the following primary antibodies: mouse-anti-human CD20 (Clone L26, Biocare), mouse-anti-CD68 (KP1; Biocare), rabbit anti-CD20 (Polyclonal, Thermo Scientific), rabbit anti-CD3 (SP7; Labvision/Thermo Scientific), rabbit anti-CD4 (EPR6855; Abcam), or rabbit anti-Ki67 (Clone SP6, Invitrogen/Thermo Scientific). Goat secondary antibodies (Jackson Immunoresearch Laboratories) against mouse, rabbit, or human IgM conjugated to Alexa Fluor 488, Alexa Fluor 647 or Cy5 were used. Sections were counterstained with 1μg/mL DAPI and mounted in Prolong Gold (ThermoFisher Scientific).

## Duplex in situ hybridization combined with immunofluorescence

For simultaneous visualization of both SIV vRNA and CAR/CXCR5-transduced cells, the RNAScope multiplex fluorescent kit V2 (Advanced Cell Diagnostics) was used with the opal fluorophores system (Akoya Bioscience) according to the manufacturer's instructions and as described previously [66] with some modifications. In brief, 5μm FFPE tissue sections on slides were deparaffinized as described above. Sections were pretreated with H$_2$O$_2$ (to block endogenous peroxidase activity) and washed in dH$_2$O. Heat-induced epitope retrieval was achieved by boiling sections in RNAScope 1× target retrieval buffer (Advanced Cell Diagnostics). Sections were washed, dehydrated in absolute ethanol, and air-dried. Sections were incubated with protease solution, rinsed twice in dH$_2$O and incubated with pre-warmed premixed target probes (all from Advanced Cell Diagnostics) in which SIVmac239 no env antisense probe channel 2 (C2) was diluted in the custom-made probe for the gammaretroviral vector to detect the CAR/CXCR5-transduced cells channel 1 (C1) at C2: C1 1:50 ratio overnight at 40°C.

Sections were washed with a 0.5× RNAScope wash buffer. Amplification and HRP-C1 and HRP-C2 signal development were performed as recommended by the manufacturer with the modification of the use of 0.5× RNAScope wash buffer instead of 1× RNAScope wash buffer and use of a 1:150 dilution of Opals (all from Akoya Bioscience) instead of 1:1500. For ileum sections, a 1:100 dilution of all Opal dyes was used. Opal 570 and Opal 690 were used for C1 and C2, respectively. For immunofluorescence staining, sections were washed twice in TBST (TBS- tween 20–0.05% v/v), blocked in 10% NGS- TBS-1% BSA and incubated with primary antibodies diluted in TBS-1% BSA for 1 h at RT. Primary antibodies included the same antibodies described in Singleplex vRNA in situ hybridization combined with immunofluorescence. The sections were washed and incubated with secondary antibodies, Opal Polymer HRP Ms + Rb for 10 min at RT. After washing, the sections were incubated with Opal 520 diluted 1:150 in the multiplex TSA buffer (Advanced Cell Diagnostics) for 10 min at RT. After washing, sections were counterstained with 1μg/mL DAPI and mounted in Prolong Gold (ThermoFisher Scientific).

## Quantitative image analysis for RNAScope

Sections were imaged using a Leica DM6000 confocal microscope. Montage images of multiple $512 \times 512$ pixels were created and used for analysis. F and EF areas were delineated using Leica software with B cell follicle areas identified morphologically as clusters of closely aggregated brightly stained $CD20^+$ or $IgM^+$ cells. Some sections were co-stained with goat anti-human IgM-AF647 (Jackson ImmunoResearch) and mouse-anti-human CD20 antibodies (clone L26, Biocare Medical, Inc.) to confirm that both antibodies co-localized similarly in B cell follicles. Cell counts were done using LAS X (Leica confocal) software; each cell was demarcated using a Leica software tool to avoid counting the same cell twice. Leica software was used to measure the delineated areas for cell counts. To determine the percentage of follicles that had CAR/CXCR5-T cells over time post-infusion, a total of 790 follicles were evaluated for presence of CAR/CXCR5-T cells with a median of 302 follicles (range, 172–316) per animal. To determine the levels of CAR/CXCR5-T cells/$mm^2$ in follicular areas, over time post-infusion, a median of 8.4 $mm^2$ (range, 6.8–8.9 $mm^2$) of follicular area was analyzed with a total of 172 follicles analyzed with a median of 57 follicles (range, 48–67) per animal. In addition, a total of 190 follicles were examined to determine the percentage of follicles that had a cluster of expanding CAR/CXCR5-T cells at the edge of the follicle at 2 DPT, with a median of 95 follicles (range 90–100). To determine the percentage of follicles with free virions bound by FDC over time post-infusion, a total of 518 follicles with a median of 146 follicles (range 140–232) per animal. To determine the levels of SIV $RNA^+$ cells/$mm^2$ in follicular areas, over time post-infusion, a median of 6.46 $mm^2$ (range 5.48–6.84 $mm^2$) of follicular area was analyzed with a total of 131 follicles analyzed with a median of 45 follicles (range 38–48) per animal. To determine levels of CAR/CXCR5- T cells/$mm^2$ and level of SIV $RNA^+$ cells/$mm^2$ in treated animals, a median of 19.9 $mm^2$ (range of 15.7–34 $mm^2$) of EF areas were analyzed per animal. To confirm the specificity of the custom-made probe that we designed to detect the gammaretroviral CAR/CXCR5 construct and to determine the level of SIV $vRNA^+$ cells in F areas, LN tissues from three untreated control animals were hybridized to the custom made probe and an SIV probe; a median of 2.36 $mm^2$ (range 0.79–2.99 $mm^2$) of follicular area was analyzed with a total of 53 follicles analyzed with a median of 22 follicles (range 5–26) per animal. To determine the level of SIV $RNA^+$ cells/$mm^2$ in untreated control animals, a median of 3.6 $mm^2$ (range of 0.68–9.3 $mm^2$) of EF areas was analyzed per animal. Effector SIV-specific CAR/CXCR5-T cells to SIV $vRNA^+$ target cell (E: T) ratios in secondary lymphoid tissues were calculated by dividing the levels of CAR/CXCR5-T cells /$mm^2$ by the level of SIV $vRNA^+$ cells/$mm^2$ within a specific tissue area.

For these analyses, for each animal we examined 2 to 3 tissue sections, with at least 6 follicles (range 6–9) examined per animal. To determine the range of E: T ratios, when the denominator was 0, we used the value of the numerator. When the numerator was zero, 1 was used as the numerator. For determination of the distance between follicular SIV vRNA$^+$ cells and the nearest CAR/CXCR5-T cell, for R14025, 3–9 follicles were examined from each of 3 spleen sections. For the T2 animals, all follicular SIV vRNA$^+$ cells were examined in 2 sections (8–28 follicles/section) for each animal.

### Immunohistochemistry and analyses

Indirect immunohistochemistry and in situ MHC-tetramer staining was performed on fresh tissue specimens shipped overnight, sectioned with a compresstome[106] and stained as described previously [107–109]. Briefly, sections were stained with 0.4 μg/mL rabbit -anti-human CD20 polyclonal antibodies (Neomarkers) and 2 μg/mL rat-anti-human CD3 antibodies (clone MCA1477, BioRad). Then, sections were stained with secondary antibodies by incubating with 0.3 μg/mL Alexa Fluor 488-conjugated goat-anti-rabbit antibodies, and 0.2–0.3 μg/mL Cy5-conjugated goat anti-rat antibodies overnight at 4˚C. Secondary antibodies were obtained from Jackson ImmunoResearch Laboratories (West Grove, PA). Sections were imaged using a Leica DM6000 confocal microscope. Montage images of multiple 512 × 512 pixels were created and used for analysis. Confocal z-series were collected in a step size of 3 μm. Images were opened and analyzed in LAS X (Leica confocal) software directly. We used the LAS X software to create montages of multiple projected confocal serial z-scans. Follicular areas were identified morphologically as clusters of brightly stained, closely aggregated CD20$^+$ cells. F and EF areas were delineated and measured using LAS X software. Areas were not included if they showed loosely aggregated B cells that were ambiguous. To prevent bias, the yellow CTV channel was turned off when F and EF areas were delineated. Cell counts were performed on single z-scans.

### Statistical analyses

Statistical analyses utilized GraphPad Prism 8.3.0 for Windows (GraphPad Software, San Diego, CA). Specific tests are indicated in the figure legends. Correlations were determined using Spearman's correlation, assuming independence.

## Supporting information

**S1 Fig. CAR/CXCR5 T cells within lymphoid aggregates in the ileum.** Representative image of ileum tissue from R14025 (T0), showing CAR/CXCR5-T cells (red) and SIV vRNA (white) cells detected by RNAScope ISH. The right panel is an enlargement from the left panel showing CAR/CXCR5 T cells and SIV vRNA$^+$ cells in a lymphoid aggregate that is likely a Peyer's patch, delineated by anti-CD20 staining (green). Scale bar is 500 μm for the left panel and 50 μm for the right panel.
(TIF)

**S2 Fig. Immunotherapeutic cell infusion leads to transient increases in cytokine levels and cell accumulation in bronchoalveolar lavage fluid.** Serum samples from all treated and control animals were analyzed for post-infusion production of cytokines using a non-human primate (NHP)-specific (A) tumor necrosis factor (TNF) alpha, (B) interferon (IFN) gamma, (C) interleukin (IL)-6 and (D) IL-2 multiplex Luminex assay. Each point represents the average of two determinations with error bars representing the standard deviation. The limit of detection for each assay is indicated by the dashed line. Most determinations were below the limit of

detection. Lung accumulation of CAR T cells was determined by analysis of bronchoalveolar lavage (BAL) samples. Cells were isolated from BAL and analyzed for (E) the percentage of CD4-MBL CAR$^+$ cells in the CD3$^+$ T population, for all treated and control animals, by flow cytometry or (F), for T2 animals only, the number of copies of CAR/cell in the total cell population by quantitative real-time PCR.
(TIF)

**S3 Fig. Representative flow plots from cells prepared for infusion.** $1–2 \times 10^6$ cells were stained with the antibodies listed in the Flow Cytometry section of Materials and Methods. (A) The gating strategy for determination of co-expression of CAR (MBL) and CXCR5 in the infused T cell product. The CD8$^+$ population was used to determine central memory phenotype (CD28$^+$CD95$^+$) and CCR7 expression. Plots presented are from transduced cells infused into Rh2858 and from mock transduced cells from the same animal. (B) The gating strategy used to determine the percentage of infusion cells in PBMC in cells collected at multiple timepoints post-infusion. Plots presented are from PBMC from a treated (R2858) and a control (R12049) animal.
(TIF)

**S4 Fig. Viral levels in the cells and supernatant infused into T1 animals.** (A) Amount of gag mRNA relative to the housekeeping gene beta-actin in cell pellets using reverse transcription (RT) polymerase chain reaction (PCR). Infusion cells from the T1 animals (CAR/CXCR5) and mock transduced cells from the same animals (Mock) are presented. The bar represents the median. (B) Virus copies/ml in the supernatant of the infusion cell product for R2526 (CAR/CXCR5) as compared to the supernatant from mock transduced cells from the same animal (Mock) and the PBS/10% autologous serum used to resuspend the cells prior to transport (PBS/serum). Viral loads of the supernatant were determined by measurement of gag mRNA by reverse transcription (RT) polymerase chain reaction (PCR).
(TIF)

**S5 Fig. Levels of tetramer$^+$ CD3 T cells present in peripheral blood mononuclear cells (PBMCs).** PBMC, collected on d28 from animals in each of the three groups, were stained for Gag CM9 and analyzed by flow cytometry as described in Materials and Methods. The bar represents the median.
(TIF)

**S6 Fig. Distribution of CAR/CXCR5-T cells in the follicular and extrafollicular area of Lymph node over time post-infusion in T1 animals.** CAR/CXCR5-T cells successfully homed to the B cell follicles and persisted for up to 28 DPT in SIV-infected ART-suppressed/released animals. (A) Levels of CAR/CXCR5-T cells over time after infusion in F (green) and EF areas (blue) of LN. (B) Percentage of follicles that contained CAR/CXCR5-T cells over time post-infusion. Samples not available are marked NA.
(TIF)

**S7 Fig. Numbers of follicular CAR/CXCR5-T cells identified in situ in lymph nodes correlate with numbers of CAR-specific PBMCs identified by flow cytometry.** Correlation between follicular CAR/CXCR5-T cells/mm$^2$ by RNAScope and CD4-MBL$^+$ PBMC by flow cytometry. Association was tested using Spearman's correlation. Scales are log (value+1) on the y-axis and log (value) on the x-axis; labels use the original units. The line represents the fitted regression. Points are labeled by days post-treatment (2, 6, 14, and 28) with a unique shape for each animal.
(TIF)

## Acknowledgments

Anti-CD3 and anti-CD28 used in these studies were provided by the NIH Nonhuman Primate Reagent Resource (R24 OD010976, U24 AI126683). IL-2 used in these studies was provided by The NCI Preclinical Repository. GAG-CM9 tetramers were provided by the NIH Tetramer Core.

The authors thank the following University of Minnesota scientists: Ms. Preethi Haran for tissue staining, Mr. Matthew McMahon for tissue imaging, Ms. Chi Phan for virus preparation, Ms. Jodi Anderson, Mr. Steve Wietgrefe and Dr. Lijie Duan for assistance with RNA-Scope development, Mr. Michael Ehrhardt of the Cytokine Reference Laboratory for the Luminex assay; the following staff members at the WNPRC: Dr. Heather Simmons for pathology services, Mr. Dane Schalk for animal services, Dr. Nancy Schultz-Darken for animal project oversight; the following staff members at the University of Wisconsin, Madison: Ms. Kim Weisgrau for cell isolation and flow cytometry, Ms. Andrea Weiler for viral load determination. We also thank Dr. Jacob Estes and Dr. Kathleen Busman-Sahay of Oregon Health and Science University for assistance with RNAScope development, Dr. Mauricio Martins at the University of Miami for the donation of the chronically infected animal to the WNPRC and Dr. James Whitney of Harvard University for MHC genotyping the animals, and for the viral load data from the C and T1 group animals during early infection and ART prior to these animals being provided to our study. We also thank Natalie Coleman Fuller for assistance in editing this manuscript.

## Author Contributions

**Conceptualization:** Pamela J. Skinner.

**Data curation:** Mary S. Pampusch, Hadia M. Abdelaal.

**Formal analysis:** Mary S. Pampusch, Hadia M. Abdelaal, Emily K. Cartwright, Aaron K. Rendahl.

**Funding acquisition:** Eva G. Rakasz, Elizabeth Connick, Edward A. Berger, Pamela J. Skinner.

**Investigation:** Mary S. Pampusch, Hadia M. Abdelaal, Emily K. Cartwright, Jhomary S. Molden, Brianna C. Davey, Jordan D. Sauve.

**Methodology:** Mary S. Pampusch, Hadia M. Abdelaal.

**Project administration:** Mary S. Pampusch, Pamela J. Skinner.

**Resources:** Eva G. Rakasz, Elizabeth Connick, Edward A. Berger, Pamela J. Skinner.

**Supervision:** Eva G. Rakasz, Elizabeth Connick, Edward A. Berger, Pamela J. Skinner.

**Visualization:** Mary S. Pampusch, Hadia M. Abdelaal, Emily K. Cartwright, Jhomary S. Molden, Brianna C. Davey, Emily N. Sevcik.

**Writing – original draft:** Mary S. Pampusch, Hadia M. Abdelaal, Pamela J. Skinner.

**Writing – review & editing:** Emily K. Cartwright, Eva G. Rakasz, Elizabeth Connick, Edward A. Berger.

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
