## [Decision Letter · Decision Letter 0]

8 Sep 2021

Dear Dr Pampusch,

Thank you very much for submitting your manuscript "CAR/CXCR5-T cell immunotherapy is safe and potentially efficacious in promoting sustained remission of SIV infection" for consideration at PLOS Pathogens. As with all papers reviewed by the journal, your manuscript was reviewed by members of the editorial board and by several independent reviewers. In light of the reviews (below this email), we would like to invite the resubmission of a significantly-revised version that takes into account the reviewers' comments.

We cannot make any decision about publication until we have seen the revised manuscript and your response to the reviewers' comments. Your revised manuscript is also likely to be sent to reviewers for further evaluation.

Sincerely,

Daniel C. Douek

Associate Editor

PLOS Pathogens

Thomas Hope

Section Editor

PLOS Pathogens

Kasturi Haldar

Editor-in-Chief

PLOS Pathogens

orcid.org/0000-0001-5065-158X

Michael Malim

Editor-in-Chief

PLOS Pathogens

orcid.org/0000-0002-7699-2064

Reviewer's Responses to Questions

**Part I - Summary**

Reviewer #1: Pampusch et al present a manuscript entitled “CAR/CXCR5-T cell immunotherapy is safe and potentially efficacious in promoting sustained remission of SIV infection” for publication in PLOS Pathogens. The authors hypothesize that increased trafficking of virus-specific CD8 T cells to B cell follicles, i.e. via modification with a CD4-MBL CAR and forced expression of CXCR5, will lead to ART-free virus control in their NHP model of suppressed HIV-1 infection.

Tissue analyses in a pilot animal necropsied 2 days post-CAR infusion show predominant trafficking of the infused cells to B cell follicles; 4% of SIV+ cells in these sites were in direct contact with CAR+ cells by duplex RNAscope. A cohort of animals that received SIV-negative T cell products (T2) showed more promising impacts on post-ATI plasma viral load than animals for which infused cells were SIV+ in origin (T1). These T2 animals’ infused cells expressed Ki67 and showed dimmer CTV within B cell follicles, consistent with a proliferative phenotype at a single time point. In line with the time frame in which CAR+ cells were detected in peripheral blood (the first 2 weeks post-infusion), the vast majority of assayed follicles in T2 animals contained CAR+ signal, but CAR+ levels in blood and tissues dropped precipitously thereafter. Nevertheless, the infused cells led to particularly promising reductions in SIV viremia in the T2 cohort vs. controls.

This is a well-designed and important study that applies cutting-edge imaging assays to characterize a promising approach to HIV cure. As with the results in most preclinical HIV cure experiments, the key measure of success (ART-free virus control) is not clear-cut, although the infused CAR T cells are having an appreciable impact on viral replication in the absence of ART. The data presented here are valuable, and open numerous doors and pose important key questions for the field moving forward.

Reviewer #2: The manuscript by Pampush et al. details the use of a SIV-specific chimeric antigen receptor that is coexpressed with CXCR5 to attempt to target persistently infected cells in lymphoid follicles in SIV infected rhesus macaques. They attempt to address their central hypothesis that the “infusion of T cells engineered to express a potent SIV-specific CAR along with a B cell homing molecule, CXCR5, will control SIV-infection by reducing viral replication in follicles”. Targeting persistently infected reservoirs, and in particular, lymphoid follicles, is a key issue with many different therapeutic approaches to HIV/SIV and this study addresses a potentially unique strategy to permit virus specific CAR expressing cells to attack infected cells at this site. The rationale, strategy, and model system in this study can potentially provide unique insight into a CAR based approach to specifically target a persistently infected reservoir. This study does demonstrate, at a reasonable level for the short term, that infusion of CAR/CXCR5 expressing cells is relatively safe, as the animals did not appear to experience treatment-related toxicities. However, the data, as it is presented, falls short of justifying their conclusions and addressing their hypothesis (see below). In sum, this is a potentially interesting study attempting to target SIV/HIV persistence in lymphoid follicles that is limited by the lack of inclusion of controls demonstrating that SIV-specific CAR/CXCR5 modified cells can target this site better than unmodified cells or cells expressing the CAR alone.

**Part II – Major Issues: Key Experiments Required for Acceptance**

Reviewer #1: -In Figures 1A, 4A-B, and elsewhere, the difference between CTV and CAR/CXCR5 RNAscope labeling should be clarified. CTV will label the entire infused cell product, not just the 55.0-79.4% of cells that have been modified with the CAR/CXCR5 vector. It would be more accurate to refer to data using CTV as “the infused T cell product,” rather than CAR/CXCR5-T cells, and CTV and CAR/CXCR5-T RNAscope labeling should preferably be shown in different colors, to avoid confusion.

-What was the frequency of CAR+ (and CTV+) cells in the blood in the pilot animal in Figure 1?

-The authors state from their previous work that “the average ratio of in vivo effector (virus-specific CD8+ T cells) to target (virus-infected cells) cells is over 40-fold lower in follicular (F) compartments compared to extrafollicular (EF) compartments of secondary lymphoid tissues” (lines 68-70). Can this ratio be quantified in the current manuscript? How does it compare to the percentage of follicles with CAR+ T cells (Figure 5D) and the percent of SIV vRNA+ cells in direct contact with CAR/CXCR5-T cells (Figure 1B)?

-Are data from all 9 animals shown in each panel of Figure S1? Panel F appears to only show cohort T2, whereas panels A-C appear to only show 1-2 animals from each of the 3 cohorts. Where average values form multiple animals are shown, this should be made clearer, i.e. with a different symbol. Showing data from the control cohort in Figure S1 E-F (and/or in Figure 4E-F) would be helpful to establish the background for each assay.

-Is there direct evidence supporting “initial immediate spike in viral loads due to the presence of virus in the infused SIV-infected transduced cells” (lines 164-165)? Are data available that quantify the amount of SIV that was present in the infused cell products from cohort T1?

-Plasma viral loads at a single time point (27-30 days post-transplant, Figure 3D) are consistent with SIV RNA levels in lymph nodes at the same time point (28 days post-transplant, Figure 6), but do not assess virological trends over time. Are other data available to corroborate this finding across a broader time frame following ART interruption, e.g. plasma viral load area under the curve, PBMC-associated viral RNA, viral DNA, or SIV Gag+ cells?

-CM9 tetramer+ assays on their own seem insufficient to conclude that “the differences in viral loads between groups was not likely driven by differences in the endogenous response” (lines 181-182, Figure S3). Were other virus-specific immune response parameters measured?

-The authors highlight that their study is “the first to visually confirm the in vivo expansion of autologous SIV-specific CAR T cells” (lines 301-302), presumably referring to their imaging data. Figure S4 convincingly shows that flow and imaging readouts are tightly correlated, lending credence to their imaging approach. However, the Ki67 data in Figure 4C-D is only a single time point, which makes it difficult to draw conclusions about a kinetic property like cell expansion. Figure 5C-D does show kinetic measurements of the authors’ imaging data, but calling this “expansion” is somewhat subjective. The authors’ key assertion that they are seeing expansion without addition of exogenous antigen (line 303) necessitates a clearer rationale and/or supporting data.

Reviewer #2: There are a few critical issues that should be addressed in this manuscript:

1. A key issue with every aspect of the study in addressing their hypothesis is the lack of proper controls, ideally with animals that received non-transduced or control vector modified cells. The control animals, from all appearances, were not treated with an infused cell product. Due to this, and the reported fact that approximately 40% of the cells that were injected into the treated animals were not expressing, or were not transduced, with the CAR/CXCR5 vector, it is difficult to determine if greater levels of CAR expressing cells were infiltrating the follicles due to CXCR5 co-expression. Since the entire cell product was labeled with the fluorescent Cell Trace Violet (CTV), it is not clear whether the CAR/CXCXR5 vector modified cells were behaving in any way different than unmodified cells or infused cells in a control animal. It also is not clear, particularly in animals that received CAR/CXCR5 modification after SIV infection, whether the cells found in the follicles are endogenous SIV-specific T cells from the infused CTV-labeled fraction or are other cells that migrated there. There is some RNAscope in situ hybridization presented that shows localization of CAR/CXCR5 modified cells in and around the follicle, but it is not clear that the presence of vector expressing CXCR5 increases this. In addition, the levels of CAR/CXCR5 modified cells versus unmodified cell populations at these anatomical sites are not clear. The inclusion of a control, optimally animals receiving cells modified with a control vector, or another manner of analyzing the present data would strengthen their conclusions that the CXCR5 component on the cells was facilitating greater follicular localization.

2. The inclusion of a control along with or in the “initial” animal data shown in Figure 1 would strengthen these results and make them more meaningful. It is not clear if unmodified cells would anatomically distribute this way as well.

3. The viral load data described in Figure 3 appear to indicate possible effects of CAR/CXCR5 modified infused cells in 4 of 6 animals. However, there is natural decrease in the untreated control animals to low levels at the final time points as well. Since this control did not receive infused cells, the significance of this is not clear (other groups have demonstrated that untreated cell infusions can affect viral loads). In addition, without matched controls for the extended time period in the T2 animals shown in Figure 3c, the significance of the presence of low viral loads at these later time points is not clear. In the absence of detection of CAR/CXCR5 expressing cells, is this a natural phenomenon?

**Part III – Minor Issues: Editorial and Data Presentation Modifications**

Reviewer #1: -Were all animals A*01+? If not, could MHC haplotype contribute to virus control in any of the animals?

-Despite statements in lines 119-120, the animals in cohorts C, T1, and T2 do not appear to be particularly well-matched in terms of age, sex, or weight.

-It would be helpful to expand Figure 2 to include schematics of what tissues were collected from each animal at which time points.

- Figure 3 appears to show plasma viral loads, which makes the reference in the Figure 3 legend to a beta actin housekeeping gene and use of cell pellets confusing.

-What does the staining pattern shown in Figure S2 look like in animal(s) from the control cohort, i.e. is this flow cytometry assay specific for CAR+ cells?

-In Figures 4-5, are data available from cohort T1?

-The statement in the discussion, “Interestingly, we detected no SIV vRNA in CAR/CXCR5 cells in the examined sections.” (lines 325-326) should be tempered, indicating that only imaging-based assays were applied. Alternatively, flow- and PCR-based assays to confirm that CAR+ cells are not infected in vivo would significantly strengthen this assertion.

Reviewer #2: 1. The methodology involved in the overall approach is not clear. A better description of key elements of this, such as greater details of the CAR/CXCR5 vector, cell processing methodologies/cell activation techniques, and the inclusion or absence of conditioning (such as lymphodepletion) regimens would strengthen this manuscript. Some reference to these is provided, but a clearer description and inclusion of these in the text would strengthen this manuscript.

PLOS authors have the option to publish the peer review history of their article (what does this mean?). If published, this will include your full peer review and any attached files.

Reviewer #1: No

Reviewer #2: No
---

## [Decision Letter · Decision Letter 1]

14 Dec 2021

Dear Dr Pampusch,

Thank you very much for submitting your manuscript "CAR/CXCR5-T cell immunotherapy is safe and potentially efficacious in promoting sustained remission of SIV infection" for consideration at PLOS Pathogens. As with all papers reviewed by the journal, your manuscript was reviewed by members of the editorial board and by several independent reviewers. The reviewers appreciated the attention to an important topic. Based on the reviews, we are likely to accept this manuscript for publication, providing that you modify the manuscript according to the minor recommendations of reviewer #1.

Sincerely,

Daniel C. Douek

Associate Editor

PLOS Pathogens

Thomas Hope

Section Editor

PLOS Pathogens

Kasturi Haldar

Editor-in-Chief

PLOS Pathogens

orcid.org/0000-0001-5065-158X

Michael Malim

Editor-in-Chief

PLOS Pathogens

orcid.org/0000-0002-7699-2064

Reviewer Comments (if any, and for reference):

Reviewer's Responses to Questions

**Part I - Summary**

Reviewer #1: Pampusch et al present a revised manuscript containing important data updates. Included are new images of CAR staining in lung (Fig. 1), updated study schematics (Fig. 3), updated plasma viral load (Fig. 4) and CAR expression data (Fig. 5), imaging of CAR T cells and virus in ileum tissue (new Fig. S1), clarification of detection limits in cytokine data (Fig. S2), increased information for flow cytometry gating (Fig. S3), SIV measurements in CAR infusion products from cohort T1 (new Fig. S4), and additional quantification of imaging of CAR vs. viral RNA (new Table 3). Updated language in reference to total (CTV-labeled) vs. CAR+ T cells and clarification of important details regarding MHC-I typing, lack of comparability in animals’ demographic info, etc are appreciated. The revised interpretation of their results in the discussion section (lines 411-422) seems reasonable and is tempered.

Reviewer #2: The authors have presented a significantly revised manuscript that sufficiently addresses previous concerns. While the lack of an optimal control is not presented, they do provide additional data that strengthens their conclusions.

**Part II – Major Issues: Key Experiments Required for Acceptance**

Reviewer #1: N/A

Reviewer #2: (No Response)

**Part III – Minor Issues: Editorial and Data Presentation Modifications**

Reviewer #1: Additional imaging-based follicular E:T ratio data is a key aspect of the revised manuscript, but no methodology is presented. Important details to include in the manuscript include i) how these values were calculated, ii) the maximum distance between a given CAR RNA+ effector and viral RNA+ target, iii) whether this distance would be expected to facilitate direct target killing, and iv) possible limitations to calculating E:T ratios using RNA signal, e.g. due to CAR RNA+ and/or viral RNA+ cells that aren’t expressing CAR/virus proteins.

Reviewer #2: (No Response)

PLOS authors have the option to publish the peer review history of their article (what does this mean?). If published, this will include your full peer review and any attached files.

Reviewer #1: No

Reviewer #2: No

Figure Files:

Data Requirements:

Reproducibility:

References:

---

## [Editor Report · Decision Letter 2]

18 Jan 2022

Dear Dr Pampusch,

We are pleased to inform you that your manuscript 'CAR/CXCR5-T cell immunotherapy is safe and potentially efficacious in promoting sustained remission of SIV infection' has been provisionally accepted for publication in PLOS Pathogens.

Best regards,

Daniel C. Douek

Associate Editor

PLOS Pathogens

Thomas Hope

Section Editor

PLOS Pathogens

Kasturi Haldar

Editor-in-Chief

PLOS Pathogens

orcid.org/0000-0001-5065-158X

Michael Malim

Editor-in-Chief

PLOS Pathogens

orcid.org/0000-0002-7699-2064
---

## [Editor Report · Acceptance letter]

2 Feb 2022

Dear Dr Pampusch,

We are delighted to inform you that your manuscript, "CAR/CXCR5-T cell immunotherapy is safe and potentially efficacious in promoting sustained remission of SIV infection," has been formally accepted for publication in PLOS Pathogens.

Best regards,

Kasturi Haldar

Editor-in-Chief

PLOS Pathogens

orcid.org/0000-0001-5065-158X

Michael Malim

Editor-in-Chief

PLOS Pathogens

orcid.org/0000-0002-7699-2064